# Characterization of atmospheric methane release in the outer Mackenzie River Delta from biogenic and thermogenic sources.

Daniel Wesley[1,2], Scott Dallimore[3], Roger MacLeod[3], Torsten Sachs[4], David Risk[1]

[1]Earth and Environmental Sciences Department, St. Francis Xavier University, Antigonish, B2G 2W5, Canada
[2]Department of Environmental Science, Memorial University, St. John's, A1C 5S7, Canada
[3]Geological Survey of Canada-Pacific, Natural Resources Canada, Sidney, V8L 4B2, Canada
[4]GFZ German Research Centre for Geosciences, Telegrafenberg, 14473 Potsdam, Germany

*Correspondence to*: Daniel Wesley (dwesley@stfx.ca)

**Abstract.** The Mackenzie River Delta is the second largest Arctic river delta in the world. Thin and destabilizing permafrost coupled with vast natural gas reserves at depth, high organic content soils, and a high proportion of wetlands create a unique ecosystem conducive to high rates of methane ($CH_4$) emission from biogenic and thermogenic sources. Hotspots are known to have a significant contribution to summertime $CH_4$ emissions in the region. Still, little research has been done to determine how often geologic or biogenic $CH_4$ contributes to hotspots in the Mackenzie River Delta. In the present study, stable carbon isotope analysis was used to identify the source of $CH_4$ at several aquatic and terrestrial sites thought to be hotspots of $CH_4$ flux to the atmosphere. Walking transects and point samples of atmospheric $CH_4$ and $CO_2$ concentrations were measured. Source stable carbon isotope ($\delta^{13}C$-$CH_4$) signatures were derived from keeling plots of point samples and ranged from -42 to -88 ‰ $\delta^{13}C$-$CH_4$, identifying both biogenic and thermogenic sources. A $CH_4$ source was determined for eight hotspots, two of which were thermogenic in origin (-42.5 ‰, -44.7‰), four were biogenic in origin (- 71.9‰ - -88.3‰), and two may have been produced by oxidation of biogenic $CH_4$ (- 53.0 ‰, -63.6 ‰), as evidenced by $\delta^{13}C$-$CH_4$ signatures. This indicates that the largest hotspots of $CH_4$ production in the Mackenzie River Delta are caused by a variety of sources.

## 1 Introduction

The Mackenzie River Delta (MRD) in the western Canadian Arctic is a unique setting for environmental methane ($CH_4$) emission to the atmosphere. Geological $CH_4$ occurs both at depth and within shallow surficial sediments, and there are many diverse settings in the area where biogenic methane production is actively occurring. Lakes, especially, in the MRD have been shown to be sources of biogenic $CH_4$ (Cunada et al., 2021; McIntosh Marcek et al., 2021). The area is characterized by a thin and destabilizing permafrost (Burn and Kokelj, 2009), high organic content soils (Schuur et al., 2008), vast amounts of deep thermogenic methane, originating from fossil hydrocarbon reservoirs (Collett and Dallimore, 1999) and over 49,000 lakes, which make up 25 - 50 % of the landscape (Emmerton et al., 2007; Lewis, 1988; Mackay, 1963). Importantly, atmospheric release of $CH_4$ during the summer in the MRD is thought to be characterized by localized areas with high methane flux, or 'hotspots' (Kohnert et al., 2017). Due to the potential for large contributions of geologic $CH_4$ and conditions where biogenic

CH$_4$ production and potential atmospheric release is likely to occur, it is important to understand these sources of CH$_4$ emissions, especially in areas of high-rate emissions.

Studies of atmospheric methane flux in the Arctic suggest that there are several factors that can influence methane dynamics in terrestrial and aquatic settings. These include environmental controls such as vegetation type, oxygen availability, soil moisture and soil temperature (including active layer regime) which can affect CH$_4$ and CO$_2$ production, oxidation, transport, and emissions (Rawlins et al., 2010; Treat et al., 2018). In addition, ecosystem heterogeneity can cause large variations in these environmental controls, all of which will be impacted by climate change (Collins et al., 2013). The MRD is characterized by a high proportion of wetlands and shallow tundra lakes (Ecosystem Classification Group, 2009, 2012;). Wetlands are considered to be the largest natural source of CH$_4$ globally, and wetland emissions are predicted to increase worldwide (Dean et al., 2018). Current global estimates of wetland CH$_4$ flux to the atmosphere range between 101-179 Tg CH$_4$/yr (Saunois et al., 2020). Non-wetland, freshwater systems are also significant contributors of CH$_4$ to the atmosphere on a global scale (Kirschke et al., 2013) which is estimated between 117-212 Tg CH$_4$/yr (Saunois et al., 2020). Ebullition from sediments is one path for CH$_4$ to enter the atmosphere from freshwater bodies (Saunois et al., 2016). In fact, a recent study showed that despite substantial winter-derived CH$_4$ being retained in the bottom waters of a lake in the MRD due to ice cover, CH$_4$ migrated to the atmosphere during the open water period (McIntosh Marcek et al., 2021). In the MRD, Arctic CH$_4$ the frequency of CH$_4$ hotspots decreases exponentially as distance to standing water increases (Elder et al., 2020; Baskaran et al., 2022). Emissions from thermokarst water bodies, such as those in the MRD, are expected to increase in the future due to longer annual ice-free periods (Wik et al., 2016; Marsh, 1990). Moreover, lakes (Kohnert et al., 2018), and natural seeps of thermogenic CH$_4$ (Bowen et al., 2008; Kohnert et al., 2017) are known sources of CH$_4$ in the MRD.

Hodson et al. (2020) found that six pingos in Svalbard had a range of annual flux rates between 76.4 and 364 kg CH$_4$/year and concluded that pingos require further study due to their potential contribution of CH$_4$ to the atmosphere. The outer MRD and the Pleistocene deposits of the Tuktoyaktuk Peninsula are home to about 2363 pingos (Wolfe et al., 2021). Release of CH$_4$ from pingos in the region could represent a significant, unaccounted source of CH$_4$ to the atmosphere making it a critical area for further study of pingos as a source of CH$_4$. The migration of CH$_4$ to the atmosphere from pingos is still poorly understood and additional studies of CH$_4$ production from pingos will help to improve Arctic CH$_4$ emission estimates.

Determining the ratios of $^{13}$C/$^{12}$C or stable carbon isotope ratio ($\delta^{13}$C-CH$_4$) is one of the best established methods to assess CH$_4$ sources to the atmosphere. Globally, atmospheric CH$_4$ has a background stable carbon isotope ratio of approximately -47 ‰ (Allan et al., 2001; Nisbet et al., 2016). Biogenic sources are depleted in $^{13}$C and therefore have a lower stable carbon isotope ratio, whereas thermogenic sources are enriched in $^{13}$C and have a higher $\delta^{13}$C-CH$_4$ (Brownlow et al., 2017). However, thermogenic signatures in particular can vary significantly, even within the same field, as they are influenced by the source rocks and formation processes (Schoell, 1980). While there have been numerous hydrocarbon exploration wells drilled in the MRD area, reported thermogenic $\delta^{13}$C-CH$_4$ ratios in the MRD are very limited. Collett and Dallimore, (1999) reported a value of -50‰ from below ice bonded permafrost from a well in the study area. Consistent with Collett and Dallimore, (1999), we consider isotopic values > -50 ‰ to indicate thermogenic sources while values < -70 ‰, indicate

biogenic $CH_4$. Intermediate values may result from the oxidation of biogenic $CH_4$ or from gas which contains a mixture of biogenic and thermogenic $CH_4$. Oxidation of $CH_4$ can occur if gas migrates through an oxidizing environment such as the aerobic zone of the soil or a wetland. This can result in a higher $\delta^{13}C$-$CH_4$ signature due to a preference for bacteria to oxidize $CH_4$ containing the lighter isotope ($^{12}C$), enriching the remaining $CH_4$ with $^{13}C$ (Chanton et al., 2005).

Migration of $CH_4$ through discontinuities in the permafrost is common in regions with thin permafrost similar to the
MRD as well as production in the organic-rich active layer during anoxic conditions (Barbier et al., 2012; Liebner and Wagner, 2007; Lupascu et al., 2012). Previous work has shown that thermogenic (Bowen et al., 2008; Walter Anthony et al., 2012; Walter et al., 2006) and biogenic (Zona et al., 2016; Walter Anthony et al., 2010) hotspots are present in Arctic permafrost environments, and suggest that both are present in the outer MRD (Kohnert et al., 2018). The Arctic is experiencing rapid climate change where soil temperatures are increasing in the outer MRD (Burn and Kokelj, 2009), and permafrost is warming
both in the Canadian Arctic (Farquharson et al., 2019; Mamet et al., 2017) and globally (Biskaborn et al., 2019). Elder et al. (2021) observed diffusive flux averaging 48.75 mg $CH_4$ $m^{-2}$ $hr^{-1}$ and peaking at 1,008 mg $CH_4$ $m^{-2}$ $hr^{-1}$ directly over thawed permafrost on the edge of a thermokarst lake in interior Alaska. Talik formation is common below lakes in the MRD (Burn, 2002) and will increase with permafrost thaw. Terrestrial thermokarst hotspots are estimated to account for roughly 4% of the pan-Arctic $CH_4$ budget but make up only 0.01% of the northern permafrost land area (Elder et al., 2021). This will provide the
means for greater $CH_4$ release from thermogenic and biogenic sources in the MRD in the future.

Due to the varied thermogenic and biogenic $CH_4$ sources in the MRD, it is important to determine the contribution of each source to the atmosphere as a basis for appraising carbon budgets. A lack of understanding of $CH_4$ sources in the MRD could lead to an underestimation of permafrost greenhouse gas emissions in the region and assessments of changes that could occur from ongoing and future climate change. To date, very little research has been done to appraise geologic vs biogenic
$CH_4$ contributions both at a regional scale and for hotspots with concentrated atmospheric flux. A recent study by Kohnert et al. (2017) found that about 1% of the mapped area in the outer MRD was an extremely high source of $CH_4$ with flux rates above 5 mg $m^{-2}$ $h^{-1}$. These authors assumed that these hotspots were primarily of geologic origin ($CH_4$ produced beneath the permafrost, including thermogenic $CH_4$ from natural gas reserves that has the potential to migrate through discontinuities in the permafrost) since the inferred flux rates of the hotspots identified were significantly greater than the maximum values from
studies published prior to 2017 of around 4 mg $CH_4$ $m^{-2}$ $h^{-1}$ detected for biogenic fluxes north of 61°N (Friborg et al., 2000; Sturtevant et al., 2012; Sachs et al., 2008). These fluxes also occurred in the summer period when most lakes were fully oxygenated, reducing biogenic emissions. According to a recent meta-analysis, the cut off value of 4 mg CH4 $m^{-2}$ $hr^{-1}$ used by (Kohnert et al., 2017) is approximately double the mean flux rate for Arctic and boreal regions (Kuhn et al., 2021). More recent work has shown that exceptionally high flux rates averaging 48.75 mg $CH_4$ $m^{-2}$ $hr^{-1}$ can be attributed to biogenic production,
with a stable carbon isotope signature of -73.8 ‰ (Elder et al., 2021; Hasson, 2022). Importantly, isotopic signatures have not been extensively used to determine the source of atmospheric $CH_4$ at hotspots in the outer MRD. These sources could behave differently than the current understanding of the region and other, similar Arctic environments.

The goal of this study was to undertake a first appraisal of the source of $CH_4$ from hotspots in the MRD with varied geology and permafrost conditions as identified by Kohnert et al (2017). This research objective was addressed by measuring the stable carbon isotope ratio of atmospheric methane emissions to determine if they were from possible thermogenic or biogenic sources. We hypothesised that the largest hotspots in the MRD include contributions of biogenic $CH_4$ due to the abundance of environmental settings where modern methane is being produced (Cunada et al., 2021).

## 2 Setting

The MRD is the second-largest Arctic river delta in the world and the largest river delta in Canada (Walker, 1998). It occurs between the higher elevation Pleistocene deposits of the Yukon coastal plain to the west and the Tuktoyaktuk coastlands to the east. The subaerial delta is thought to have formed in a glacial valley filled with late Pleistocene glacial sediments that were subsequently overlain by Holocene-aged deltaic deposits (Hill et al., 2001). A succession of Tertiary-aged hydrocarbon bearing sediments occurs at depth beneath the entire MRD region, with a number of large thermogenic hydrocarbon fields having been identified by industry (Osadetz and Chen, 2010). Deltaic sediments occurring in the near-surface consist mainly of fine sand and coarse silts which are >100 m thick in most of the MRD but can be less than 20 m thick in the extreme north-eastern areas (Dallimore et al, 1992; Mackay, 1963). Permafrost is considered to be continuous beneath land areas, but is largely absent beneath lakes which do not freeze to the bottom in winter (Nguyen et al., 2009). This landscape is a prime location for the formation of Arctic river taliks (Ensom et al., 2012) which can be sources of high-rate geologic $CH_4$ seeps (Sullivan et al., 2021). These river taliks can also form connecting through taliks with nearby wetlands which can create a network of discontinuous permafrost (Minsley et al., 2012). Ground ice content in deltaic sediments may exceed 50% in near-surface sediments but is substantially lower at greater depths (Collett and Dallimore, 1999; Mackay, 1963). Ground ice exists in the form of bonding cement or visible ice in excess of the pore space occurring as lenses, veins and rarely as massive ice layers (>1m in thickness).

The MRD has a variety of unique permafrost landforms including extensive areas with ice-wedge polygons and a number of isolated pingos which can range in size from just a few metres to 10-20 m high (Mackay, 1963; Wolfe et al., 2021). Bowen et al. (2008) have also identified a number of pockmark features in water bodies that are thought to be caused by geologic methane flux. The outer MRD is experiencing the ongoing effects of climate change (Burn and Kokelj, 2009), rapid coastal retreat and warming air temperatures that have risen in the past three decades at a rate that is three times the global average (GRID-Arendal, 2020).

## 2.1 Study Location

The study sites are shown in Fig. 1 and are superimposed on a map of concentrated areas with high $CH_4$ flux rates derived from published results by Kohnert et al. (2017). Eight of the nine study sites were located within the MRD and one site was located within the Tuktoyaktuk coastlands. Five study sites were located at large (about 1-5 km$^2$) but well-defined hotspots

with methane flux rates, determined from aerial surveys by Kohnert et al. (2017) to be in excess of 5 mg hr$^{-1}$m$^{-2}$ (Pingo 1,

Pingo 2, Wetland 2, Wetland 3, Site 9). Site 9 was a tundra site vegetated mainly with shrub willows and alders. Walking transects were conducted at all of these sites due to their large and areal nature in order to increase sampling coverage. Discrete point samples were taken at each site for stable carbon isotope ratio ($\delta^{13}$C-CH$_4$) determination.

Four additional sites (Pingo 3, Wetland 1, Lake 1, Channel Seep) were at locations where point source, aquatic seeps where concentrated ebullitions of CH$_4$ flux were seen in open water in the summer, or in holes in newly formed ice in the fall.

Channel Seep and Lake 1 were previously known to researchers in the field party, while Pingo 3 and Wetland 1 sites were identified by holes in the ice which were observed from the helicopter while passing overhead during the fall. Discrete point samples were taken for stable carbon isotope ratio ($\delta^{13}$C-CH$_4$) analysis. Photographs of each site and descriptions are included in Fig. 2.

The main study sites were situated in the lower subaerial delta plain, which consists of deltaic sediments, many

meandering river channels and numerous thermokarst lakes (Burn and Kokelj, 2009). The permafrost in this area is typically less than 100 m thick and continuous beneath land areas (Nguyen et al., 2009; Hu et al., 2014). However, taliks, or thawed zones in the permafrost, form below most lakes and channels and often penetrate the entire permafrost interval (Burn, 2005). The high number of lakes in the area is characteristic of many Arctic and Subarctic deltas (Marsh, 1990). In the outer MRD, lakes tend to remain oxygenated and have well-established macrophyte communities by the end of the summer (McIntosh

Marcek et al., 2021). Many of the lakes are isolated from the flow of the channels of the Mackenzie River, except during storm surge inundation (Marsh, 1990). Terrestrial areas are dominated by mixed tundra vegetation with some areas (particularly along the edge of river channels) with well-developed shrub willow and alders, however, other areas are more sparsely vegetated with exposed delta muds and sedge vegetation (Burn and Kokelj, 2009; Gill, 1972). Many flat-lying terrestrial areas are covered by 10-40 cm of standing water during late summer. Since all sites were only accessible by helicopter, field access

was largely dependent on weather and ground conditions at each site.

Pingo 3 is located to the east of the MRD in an area referred to as the Tuktoyaktuk coastlands (Ecosystem Classification Group, 2012). This site was sampled in the fall in a shallow pond which formed in the crater of a collapsed pingo. Steady ebullition of CH$_4$ occurring in the pond maintained open holes in newly formed ice. This site is located east of the airborne eddy covariance flux surveys conducted by Kohnert et al. (2017) (Fig. 1). The surficial materials in this area are

characterized by glacial moraine with continuous shrub tundra vegetation. However, the pingo itself was located in a drained lacustrine basin and similar to many other pingos in the Tuktoyaktuk coastlands area in general (Ecosystem Classification Group, 2012). The permafrost of the Tuktoyaktuk Coastlands typically is more than 400 m thick and therefore through going taliks are much more rare (Hu et al., 2014).

## 3 Methods

In the present study, we investigated several aquatic and terrestrial hotspots of atmospheric $CH_4$ flux. Study sites visited in the summer were chosen based on areas with high methane flux rates determined from airborne eddy covariance flux surveys, conducted in the outer MRD by the German Research Centre for the Geosciences (GFZ) together with the Alfred Wegener Institute for Polar and Marine Research (AWI) (Kohnert et al., 2017). We conducted ground sampling surveys in July 2019 and 2021. The airborne eddy covariance flux surveys were conducted seven years prior, during the same month in 2012 and

2013. We also carried out fieldwork in October of 2019 focusing on sampling near holes in the ice forming on water bodies caused by high rate $CH_4$ ebullition.

### 3.2 Sample Collection and Analysis

Geolocated air samples were collected in the field and subsequently analysed at the Aurora Research Institute laboratory in Inuvik, Northwest Territories, Canada or at St. Francis Xavier University in Antigonish, Nova Scotia, Canada. $CH_4$ and $CO_2$

concentrations and $\delta^{13}C$-$CH_4$ were determined using a Picarro G2210i analyser. The Picarro G2210i analyser is accurate to within 1 ‰ $\delta^{13}C$-$CH_4$, 0.001 ppm $CH_4$, and 1 ppm $CO_2$. All point samples were analysed for at least 5 minutes and the values averaged, which increases the Picarro G2210i analyser accuracy to 0.1 ‰ $\delta^{13}C$-$CH_4$, 0.0001 ppm $CH_4$ and 0.01 ppm $CO_2$. Only point samples were used for the determination of $\delta^{13}C$-$CH_4$ source signatures. Sample positions were recorded with a Garmin eTrex 10 handheld GPS, which has an accuracy of +/- 3.65 m. Gas mixing ratios were standardized to Ameriflux

FB04306 breathing grade air (benchmarked to 0.5 ppb) Five walking transects of atmospheric $CH_4$, and $CO_2$ concentrations were completed at Pingo 1, Pingo 2, Wetland 2, Wetland 3, and Site 9 in 2019 (Fig. 3). Hotspots determined by airborne eddy covariance (Kohnert et al., 2017) were typically several square kilometres in extent and therefore it was impractical to sample the entire feature. Sampling transect locations were selected within the general hotspot area where the airborne eddy covariance flux rates were highest. Samples were concentrated around features such as wetland areas or pingos that were considered

possible sources of $CH_4$. Unfortunately, this has the potential to bias sampling to the $CH_4$ produced in and around these features. Walking transects were carried out by filling a 30 m coil of 4 mm inside diameter aluminum Synflex tubing while walking at a steady pace across the ground. Walking transects covered a distance between 600-800 m and took approximately 20 minutes to fill a single coil. Coils of tubing were purged with nitrogen and sealed prior to sampling. A constant flow rate of 20 standard cubic centimetres per minute (CCM) was maintained by attaching a small pump and a flow controller to the coil of tubing.

Walking transect air samples were analysed immediately on return from the field site by feeding the sample into a Picarro G2210i analyser at the same rate at which it was filled. Concentrations of $CH_4$ and $CO_2$ were measured every 1-2 seconds as the sample was being analysed. Mixing of the air sample inside the tube between collection and analysis is limited due to the small diameter of the tubing. A similar method was used during drone-based $CH_4$ measurements (Andersen et al., 2018). Two walking transects were collected at Pingo 3 and Wetland 1 in 2021 using a Li-Cor LI-7810 gas analyser over an approximately

10 minute period and covered a distance of approximately 200-300 m. The Li-Cor LI-7810 gas analyser has an accuracy of

0.25 ppb. Sample air was drawn from approximately 1 m above ground level. Each walking transect location was selected by completing one transect upwind and one downwind of the estimated location of the highest flux concentration in order to obtain background concentrations at each site. If features that represented potential sources such as pingos or wetlands were present, then walking transects were taken upwind and downwind of the feature. Discrete point samples were taken parallel to each walking transect, 3-5 m further away from the centre of the hotspot. This sampling method was designed to cover the most area, with the fewest samples, in the shortest amount of time possible. By using this method we were still only able to cover a small fraction of the hotspot (5-10% by area). By sampling near observed potential sources or near the zone of the highest flux rate, we increased the likelihood of pinpointing the source.

Discrete point samples were taken at all sites by filling 1 L Tedlar bags with ambient air from approximately 1 m above ground level. At aerial eddy covariance sites point samples were taken along each walking transect. The point samples were used to measure stable carbon isotope ratios ($\delta^{13}$C-CH$_4$). Point samples at aquatic seep sites were taken at the seep location and additional samples were taken up to 5 m away from known sources in order to measure source and background CH$_4$ concentrations. At Pingo 3 and Wetland 1 point samples were taken directly over holes in the ice by throwing a buoy with tubing attached to it onto the water in the open hole in the ice and using a pump to draw air through the tubing into a sample bag. At Lake 1 samples were collected from a boat which positioned the sample inlet directly above any discovered seeps. At Channel Seep samples were collected by extending a pole with the sample inlet attached to the end out over the seep from the channel bank. All samples were taken from the air less than 10 cm from the seep with the exception of the channel seep sample which was approximately 2 m from the seep. Walking transects and discrete point samples were completed at Pingo 3 and Wetland 1 in the summer of 2021 to compare overall site variation to point source samples Wind regime at sample sites was measured with a Kestrel 1000 handheld anemometer which has an accuracy of +/- 0.1 m/s.

### 3.3 Determination of CH$_4$ Source Stable Carbon Isotope Value ($\delta^{13}$C-CH$_4$)

Keeling plot analysis of the discrete point samples was used to determine the stable carbon isotope signature of the CH$_4$ source at each site. Keeling plot analysis is a common approach used to determine a source of carbon entering the atmosphere by measuring the change in $\delta^{13}$C-CH$_4$, or fractionation, that occurs as more carbon from that source is added (Kohler et al., 2006). This analysis uses a mass balance approach and takes into account the relative difference in stable carbon isotope ratios of the atmosphere and an additional carbon source (Kohler et al., 2006; Pataki et al., 2003). The Y-intercept of a linear regression of the $\delta^{13}$C-CH$_4$ vs. the inverse of the CH$_4$ concentration will indicate the source which contributed to the increase in atmospheric CH$_4$. This approach was first used by Keeling (1958, 1961) to determine the source of CO$_2$ contributions to the atmosphere (Pataki et al., 2003). One main assumption of Keeling plot analysis is that there are only two components being measured, the source being released at the surface/atmosphere interface and the background regional atmospheric signature (Pataki et al., 2003). This assumption is challenging to achieve under field conditions as there can be multiple potential sources of CH$_4$ if the sampling is carried out over a broad area or in windy conditions that may cause mixing. For the walking transects described in our study, we accepted this limitation as we were attempting to appraise rather large hotspots identified by Kohnert et al.

(2017). On this basis, it seemed reasonable to accept that our atmospheric point samples, conducted within the assumed source

of these six hotspots, could be representative of a blended $\delta^{13}$C-CH$_4$ signature as might be measured by these researchers at the elevation that the survey aircraft was flying (40-80 m above ground level). In comparison, at sites where ebullition was observed and there was a known point source of emission, samples could be collected directly over the source. This enables a high degree of certainty that only one source is being measured at Pingo 3, Wetland 1, Lake 1, and Channel Seep sites.

## 4.0 Results

Data obtained from walking transects at each site along with wind speed are shown in Table 1. Atmospheric CH$_4$ concentrations were elevated at four of five airborne eddy covariance sites where walking transects were completed (Pingo 1, Pingo 2, Wetland 2, Wetland 3). Only background atmospheric concentrations of CH$_4$ were observed at Site 9 (Table S1), therefore, it was not included in the main analysis. Mean background atmospheric values recorded at the Inuvik ECCC weather station for the period of fieldwork (July 8-19, 2019) were 1.995 ppm for CH and 402 ppm CO$_2$. The maximum CH$_4$ values for walking

transects obtained were  8.734 ppm at Pingo 2, 6.135 ppm  at Pingo 1, 2.264 ppm at Wetland 3, 2.152 ppm at Wetland 2, and background values at Site 9. During the walking transects concentrations above 2.5 ppm were recorded for 1056 of 1850 measurements at Pingo 1 and for 285 of 2013 measurements at Pingo 2. Elevated CO$_2$ values were also observed during walking transects at two of the eddy covariance hotspot sites with values of 603 ppm at Wetland 2 and 463 ppm at Pingo 2. Site images with CH$_4$ concentrations from walking transects can be seen in Figure 3. Keeling plots of stable carbon isotope

signatures ($\delta^{13}$C-CH$_4$) are shown in Figure 4. Y-intercept values were -53.0 ‰ for Pingo 1, -63.6 ‰ for Pingo 2, -78.4‰ for Wetland 2, and -71.9 ‰ for Wetland 3.

As expected, the four sites with discrete samples in close proximity to ebullition sites had substantially elevated methane concentrations (Fig. 1). Estimates of source stable carbon isotope signatures derived from Keeling plots (Fig. 4) were -77.6‰ for Pingo 3, -42.5 ‰ for Channel Seep and -44.7 ‰ for Lake 1. Keeling plot values were -88.4‰ for Wetland 1 when

sampled in the fall (October), but -56.7 ‰ when sampled in the summer (July). Pingo 3, Lake 1, and Wetland 1 were sampled in the fall with ebullitions forming holes in the newly formed ice. Channel Seep was documented during a previous field campaign. Maximum values of 2.94 ppm CH$_4$  and 519 ppm CO$_2$ were observed at Pingo 3 and values of 12.399 ppm CH$_4$ and 488 ppm CO$_2$ were observed at Wetland 1. Fluxes were measured for CH$_4$ and CO$_2$  in 2021 at Lake 1 and Pingo 3 (Table S2).

# 5 Discussion

## 5.1 Methane characteristics observed at airborne eddy covariance hotspot sites

### Outer delta pingos

The highest $CH_4$ concentrations for walking transects co-located within airborne eddy covariance $CH_4$ hotspots were obtained in the northwestern part of the outer MRD. The elevated values measured over a dispersed area within the eddy covariance hotspots provide a basis to speculate on the possible sources for the hotspots. Estimates of source stable carbon isotope signatures ($\delta^{13}C$-$CH_4$) derived from Keeling plots were -53.0 (+/- 1.01) ‰ for Pingo 1 and -63.6 (+/- 1.87) ‰ for Pingo 2. These signatures are enriched in $^{13}C$ compared to typical biogenic sources with $\delta^{13}C$-$CH_4$ < -70 ‰ and within 14 ‰ of our assumed threshold of -50 ‰. We conclude, therefore, that the signature at these two sites could be from oxidised biogenic $CH_4$ but a geologic source for the methane for each site that is made up of a mixture of thermogenic and biogenic gas from depth, perhaps with a dominance of thermogenic methane cannot be ruled out. However, while elevated values for the walking transects were observed in close proximity to the pingo features, we note that for both sites the highest values were not on the features themselves, but a short distance away in the low-lying shrub tundra terrain surrounding the pingos.

Most pingos in the Tuktoyaktuk Peninsula generally form in areas of thick permafrost with drained lacustrine basins and are therefore assumed to be closed system pingos formed from the re-freezing of a local talik beneath the drained lakes. However, because permafrost is thin in the outer MRD, the pingos in this area can be open system pingos formed from fluid migration from beneath the permafrost interval (Mackay, 1963). We conclude that the Pingo 1 and 2 features are most likely open system pingos as they have formed in a flat-lying delta plain setting with no indication of a drained lake and nearby scientific drilling indicates only about 80 m of thermally defined permafrost and perhaps only 50-60 m of ice-bonded permafrost (Dallimore et al., 1992). Methane has been shown to occur in groundwater discharges in the vicinity of open system pingos in Svalbard, although these pingos are formed in a different geologic setting than the MRD (Hodson et al., 2019). In this case, the isotope signature of $CH_4$ in groundwater was similar to that found below the permafrost. We note a similar relationship for the pingo sites in the outer delta where scientific drilling conducted at the Unipkat well site found an isotopic $\delta^{13}C$-$CH_4$ value of -53‰ dissolved methane in core samples beneath the ice-bonded permafrost interval (Collett and Dallimore, 1999). This is an identical value to Keeling plot determinations for the Pingo 1 site (-53.0 ‰) and similar to that found at Pingo 2 (-63.6‰). A point of interest, however, is that the highest $CH_4$ concentrations at sites Pingo 1 and Pingo 2 were measured 200-400 m (Fig. 3) away from the pingos and were only as high as 2.2 ppm directly over the pingos (Table S3). One possibility for this occurrence is that the signal from the pingo itself may have been shifted by the wind, however, measured wind velocity and direction during the survey did not seem consistent with this possibility. Given that the ice bonded permafrost in the pingo itself is likely impermeable to fluid and gas flux, we conclude that the local source is likely from the terrain surrounding the pingo where indeed open system groundwater flow may be occurring. This interpretation would be consistent with recent findings in the region that water table depth, hydrology and topography are critical factors driving emissions at hotspots in the

MRD (Elder et al., 2020; Baskaran et al., 2022; Hodson et al., 2020a, b). Further research including repeat ground sampling transects and possible permafrost geophysics is warranted to further assess this hypothesis.

Collapsed Pingo 3 was formed in a much different environment than Pingo 1 and 2, in the Tuktoyaktuk coastal plain where the permafrost is > 500 m thick (Todd and Dallimore, 1998). Migration of $CH_4$ through the permafrost is much less

likely here than in the outer MRD. In this case, ebullitions of gas were emitted from a small pond with a Keeling plot stable carbon isotope source signature of -73.6 ‰ suggesting a more definitive biogenic methane source. As the permafrost is much thicker at this site, the potential for a talik penetrating the entire permafrost and providing a conduit for migration of deep thermogenic methane is much less likely.

**Wetland sites**

Two wetland sites were located at airborne eddy covariance identified hotspot sites that occurred in rather flat delta plain settings in the western part of the outer MRD (Fig. 1). These sites were dominated by low shrub tundra with some areas of exposed delta sediments with sparse sedges. We characterized these as wetlands as much of the surface had 10-30 cm of standing water. Atmospheric $CH_4$ values during walking transects at Site 9 peaked at 2.044 ppm, however, concentrations of

2.152 ppm were observed at Wetland 2 and 2.264 ppm at Wetland 3. At both sites modestly elevated values above background were found along most of the walking transects. Estimates of source stable carbon isotope signatures ($\delta^{13}C$-$CH_4$) derived from Keeling plots were -78.4‰ for Wetland 2 and -71.9 ‰ for Wetland 3. This is consistent with the knowledge that wetlands can produce significant amounts of biogenic $CH_4$ (McGuire et al., 2012; Wik et al., 2016; Andresen et al., 2017). Elevated $CO_2$ values up to 603 ppm were observed at Wetland 2. The lack of a methane signal at Site 9 and the low values at Wetlands 2 and

3 do not provide substantive validation of a possible source for the eddy covariance hotspots observed by Kohnert et al. (2017). However, it is reasonable to consider that these featureless delta plain areas of the MRD may be dominated by widely dispersed methane flux from mainly biogenic sources. Elevated $CO_2$ of 603 ppm at Wetland 2 may also support this inference since co-generation of $CH_4$ with $CO_2$ is typical of biogenic production (Chanton et al., 2005).

Discrete sampling at Wetland 1 yielded a $\delta^{13}C$-$CH_4$ Keeling plot source signature of -88.3 ‰ when sampled in

October during freeze-up and -53.4 ‰ during the summer. We conclude that this demonstrates a biogenic source during the fall since biogenic production can persist late into the cold season (Zona et al., 2016). While the sampling was carried out at the same location, methane ebullition was seen while sampling during the fall, but not during the summer. The Wetland 1 site was dominated by sedge vegetation with areas of standing water. The lack of ebullition flux at the same site during the summer and the different Keeling plot estimate suggests methane flux in this wetland setting varies seasonally. The Keeling plot source

signature of -53.4 ‰ during the summer could be caused by either oxidation of a biogenic source or contributions of both biogenic and thermogenic sources. Oxidation of $CH_4$ has been shown to be a significant source of fractionation in Arctic lakes during the summertime (Thompson et al., 2016). Groundwater inputs to lakes in permafrost areas are higher during the summer months (Olid et al., 2022), which would increase the possibility of thermogenic inputs during the summer. Seasonal shifts in

lake-produced CH₄ stable carbon isotope signatures potentially due to oxidation are known to occur but are typically observed
during winter beneath the ice cover (Michmerhuizen et al., 1996; Ettwig et al., 2016), or the transition from the ice-covered to open water periods in the spring (McIntosh Marcek et al., 2021). Similar observations for seasonal variability in terrestrial sources are not well documented in the literature, although transport of $CH_4$ from anaerobic soils with sedge vegetation has been observed to bypass the aerobic zone, limiting oxidation during the growing season (Olefeldt et al., 2013; King et al., 1998). Therefore, it is possible that there were contributions to the atmosphere from biogenic and thermogenic sources at
Wetland 1, but oxidation and varying production pathways cannot be ruled out as the reason for the signature derived during the summer sampling.

## 5.2 Methane from ebullition sources in the MRD and Tuktoyaktuk Coastlands

Some of the largest occurrences of atmospheric release of $CH_4$ in Arctic environments have been reported in association with large gas seeps of thermogenic $CH_4$, causing high-rate ebullition (Walter Anthony et al., 2012), but biogenic contributions
from microbially produced fossil $CH_4$ (Sullivan et al., 2021) and young carbon can also occur at seeps with high flux rates (Elder et al., 2018). The Channel Seep sampled in this study was at the edge of a small river channel with high-rate ebullition observed during our fieldwork. This site had a $\delta^{13}$C-CH₄ Keeling plot source signature of -42.0 ‰. Lake 1 which was conducted in the immediate vicinity of a smaller ebullition stream also had a very similar source signature of -44.7 ‰. Analyses of a seep gas sample from this lake carried out in 2008 yielded values of -290 ‰ δD-CH₄/ -45 ‰ $\delta^{13}$C-CH₄ and -230 ‰ δD-CH₄/ -37‰
$\delta^{13}$C-CH₄ (S.R. Dallimore, personal communication, January 12[th], 2023) This is well within the thermogenic isotopic field as determined by Whiticar (1999). When combined with sampling of a vigorous ebullition in a small pond by Bowen et al. (2008) with -43.3 ‰ values, three locations with similar ebullition character and isotopic values are spaced over approximately 20 km in a north-south trend. We note that the Niglingtak/Kumak hydrocarbon field, which occurs only 5 km to the east of these sites, occurs in a faulted anticline structure with the same trend. As the permafrost is less than 80 m thick in this area, it is
probable that the high ebullition sites occur where taliks have formed through the permafrost creating a migration pathway for thermogenic gasses to pass through the permafrost. In addition, a common thermogenic signature at multiple sites in close proximity but different settings suggests a possible pervasive regional source.

    The soil adjacent to Lake 1 was saturated with water, creating ideal conditions for biogenic production at the site. It is possible that we sampled multiple sources of $CH_4$ at the site; biogenic methane production in and around the lake as well as
a strong thermogenic seep. This could account for the low $r^2$ value (0.48) (Fig. 4) at Lake 1 despite highly elevated $CH_4$.

    Thermogenic seeps to the atmosphere have been documented in Arctic regions, including the Mackenzie River Delta, where thermogenic methane exists underneath the permafrost (Bowen et al., 2008; Osadetz & Chen, 2010; Walter Anthony et al., 2012). While biogenic $CH_4$ production typically results in $^{13}$CH₄ values less than -70 ‰, oxidation can result in higher $\delta^{13}$C-CH₄ signatures which are closer to that of thermogenic $CH_4$. This is due to a preference for bacteria to oxidize $CH_4$
containing the lighter isotope ($^{12}$C), enriching the remaining $CH_4$ with $^{13}$C (Chanton et al., 2005). Values of $\delta^{13}$C-CH₄ signatures as high as -44.7 ‰, which were observed at Lake 1, are not likely to be generated through oxidation of biogenic $CH_4$. Values

almost as high are exceedingly rare at these latitudes, but have been observed before, in an Arctic lake (-49.2 ‰) (Thompson et al., 2016), and in a pond formed in polygonal tundra (-44.9 ‰, -52.3 ‰) (Preuss et al., 2013; Vaughn et al., 2016). Stable carbon isotope values even higher (more enriched in $^{13}C$) than those reported in this study (-38.8 ‰) were observed below lake ice on the north slope of Alaska (Elder et al., 2018). These higher values were attributed to oxidation of biogenic $CH_4$ but, were measured in an environment where high rates of oxidation are likely. In the case of Preuss et al. (2013), almost complete oxidation of $CH_4$ to atmospheric levels was required to increase the $\delta^{13}C$-$CH_4$ signature from less than -50 ‰ up to -44.9 ‰. Additionally, these two sites are at a river channel and a lake, respectively, where oxidation would be minimal as compared to a wetland. The likelihood of the permafrost thawing completely through at these two locations is also higher than at the wetland locations, increasing the possibility of thermogenic migration.

## 6 Conclusion

To our knowledge, this is the first study to measure stable carbon isotope signatures of atmospheric methane at hotspots in the MRD. Estimates of source isotope signatures from field sites ranged from -42 ‰ to -88 ‰, indicating that the largest sites of $CH_4$ production in the MRD are caused by both biogenic and thermogenic sources. Of eight sites investigated in this study, two were thermogenic in origin, four were biogenic in origin, and two may have been produced by oxidation of biogenic $CH_4$, as evidenced by stable carbon isotope signatures and the high potential for migration of $CH_4$ from below the thin permafrost at the majority of these sites.

In this study, we were able to verify airborne eddy covariance hotspot locations using walking transects to measure atmospheric variation of $CH_4$. These methods can still be improved on as they only provide a snapshot of methane sources to the atmosphere during site visits and a true picture of annual $CH_4$ production cannot be established here. Future research should include year-round flux measurements at these sites, coupled with stable carbon isotope measurements. This would fully quantify the annual $CH_4$ emission to the atmosphere from biogenic and thermogenic sources at these sites. This study attempted to verify $CH_4$ hotspots identified from airborne eddy covariance surveys and determine their source at the same time. Because sites from remote sensing surveys were large and areal in nature, verification required covering large areas by foot. Not only is this time-consuming but can make it difficult to pinpoint exact emission locations. Combined use of portable $CH_4$ analysers with flux chamber and isotopic measurements at the locations of the highest atmospheric mixing ratios of $CH_4$ would be a more direct and methodical way separate and quantify sources, especially at sites where both biogenic and thermogenic sources are likely.

## Competing Interests

The authors declare that there are no competing interests.

**Author Contribution**

DW conceptualization, investigation, formal analysis, original draft preparation, reviewing and editing. SD funding acquisition, conceptualization, reviewing and editing. RM investigation, reviewing and editing. TS reviewing and editing. DR funding acquisition, conceptualization, reviewing and editing. All of the authors edited and approved the final version for submission.

**Acknowledgements**

We would like to acknowledge the following contributions to this work. The LiCor gas analyser used for sampling was kindly provided by Jacqueline Goordial. Support during fieldwork was provided by Jacqueline Goordial, Peter Morse and Isaac Ketchum. Samples from Lake 1 were collected by Julia Boike. Some sample analysis was provided by Laura Lapham. Aircraft support was provided by the Polar Continental Shelf Program granted to the Geological Survey of Canada. Field logistics, including lab facilities, were provided by the Aurora Research Institute. Fieldwork costs were supplemented by the Northern Scientific Training Program. The authors wish to acknowledge the advice and comments provided by the reviewers of this paper.

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

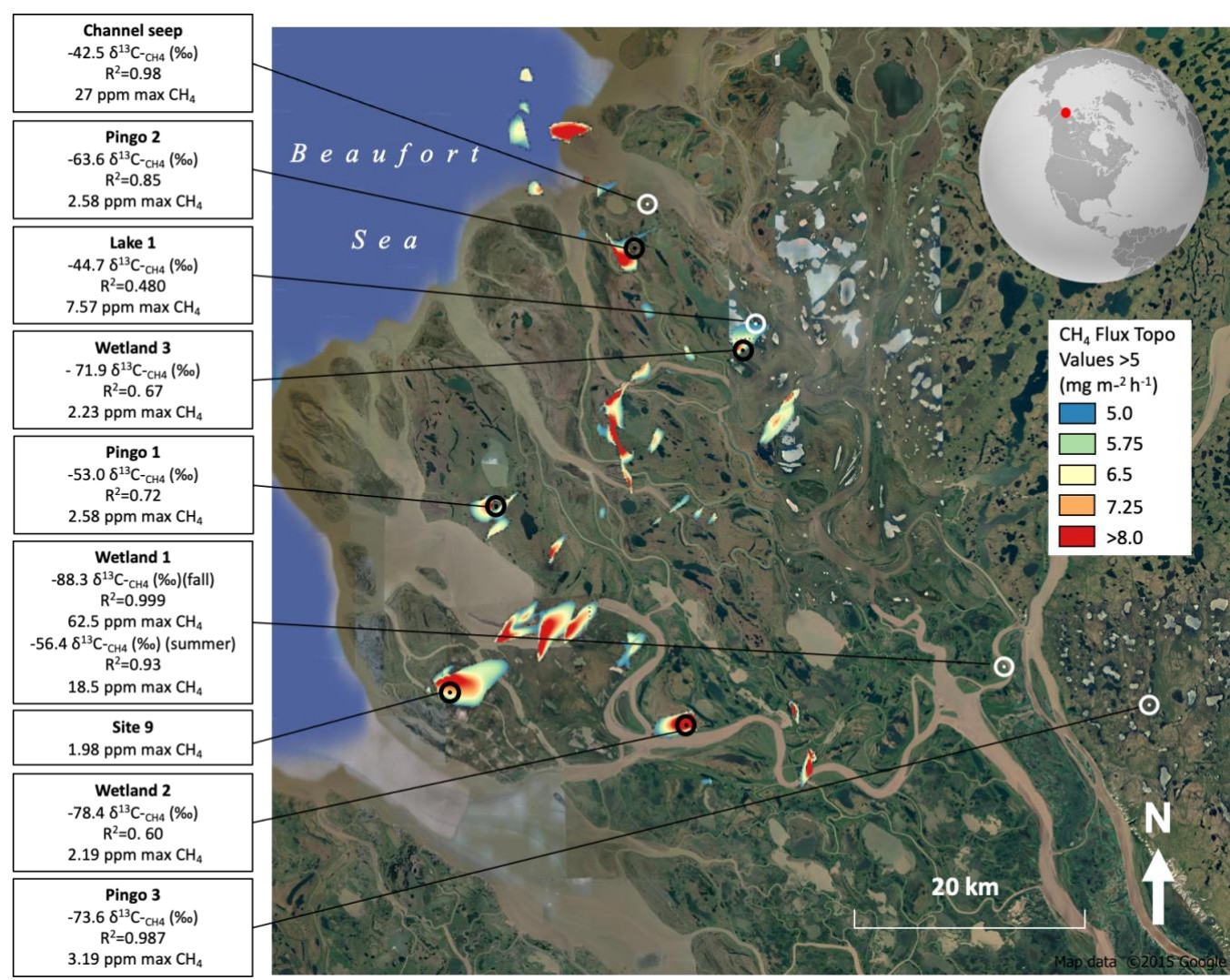

**Figure 1: Methane stable carbon isotope source signatures at sample sites in the Mackenzie River Delta. Locations are superimposed on a map of CH₄ flux rates published by Kohnert et al. (2017) Signatures varied from -42 to -88 ‰ δ¹³C-CH₄ indicating that the source of CH₄ varied at different sites and ranged from entirely thermogenic (indicated by less negative ‰ values) to entirely biogenic to a mixture of both biogenic and thermogenic. Source signatures were derived from regression of keeling plots which were based on point samples of atmospheric methane. Black symbols indicate sites identified by Kohnert et al. (2017). White symbols indicate sites identified by CH₄ ebullition.**



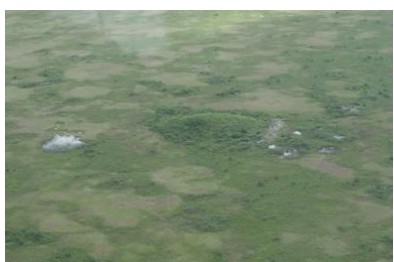

Pingo 1: A large (> 1 km wide) $CH_4$ hotspot identified by Kohnert et al. (2017). The pingo itself was approximately 95 m wide and 7.5 m high. Discrete point samples and walking transects were taken around the pingo in the area where the highest flux rates were observed by Kohnert et al. (2017).

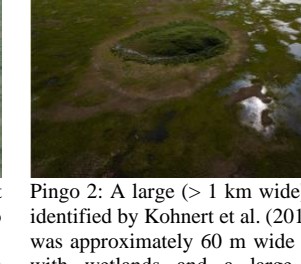

Pingo 2: A large (> 1 km wide) $CH_4$ hotspot identified by Kohnert et al. (2017). The pingo was approximately 60 m wide and 6 m high with wetlands and a large lake to the southwest. The pingo was located on the very edge of the hotspot but near where the highest flux rate was observed by Kohnert et al. (2017). Discrete point samples and walking transects were taken around the pingo and to the southwest of the pingo near the wetland.

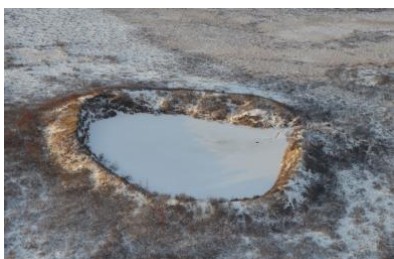

Pingo 3: A collapsed Pingo of about 80 m in diameter with a lake formed in the resulting crater. Discrete point samples were taken during the fall, close to holes in the ice and outside of the crater. Discrete point samples and walking transects were taken around the edge and outside of the crater during an additional summer field campaign. Flux chamber measurements were taken at the outlet of the lake in a sedge covered area.

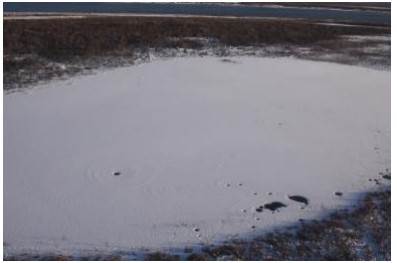

Wetland 1: A sedge dominated wetland of about 80 m in diameter. Discrete point samples were taken near holes in the ice and along the edge of the lake during the fall. Discrete point samples and walking transect were taken during an additional summer field campaign in the same location as the fall samples.

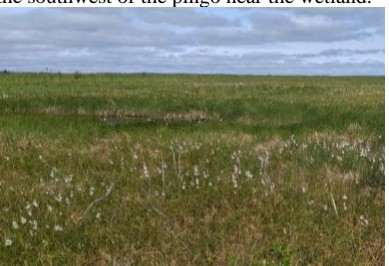

Wetland 2: A broad $CH_4$ hotspot (~ 800 m wide) identified by Kohnert et al. (2017). The site was comprised of tussock tundra dotted by many small wetlands and a few shrubs. Field of view is approximately 10 m wide. Discrete point samples and walking transects were taken where the highest flux rates were observed by Kohnert et al. (2017).

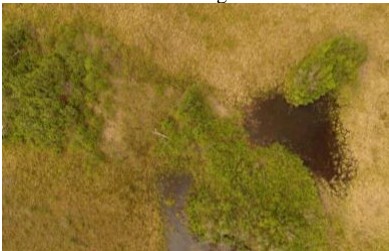

Wetland 3: A broad (~ 600 m wide) $CH_4$ hotspot identified by Kohnert et al. (2017) with mixture of wetland and grassland with patches of alders. Field of view is approximately 40 m wide. Samples and walking transects were taken where the highest flux rate was observed by Kohnert et al. (2017).

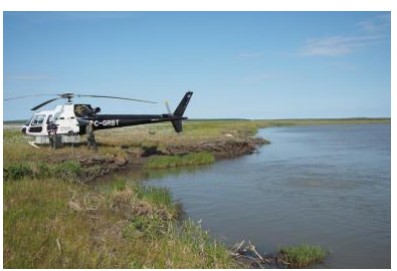

Channel Seep: A very active seep in a river channel observed during a previous field campaign. Samples were collected by extending the sample inlet on a pole over the channel and taking subsequent samples further away from the channel. No walking transects were collected at this site.

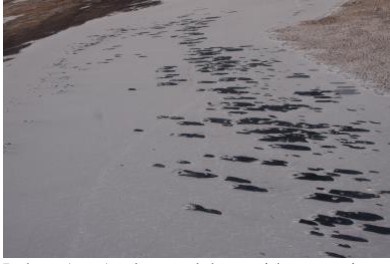

Lake 1: A large lake with prominent ebullition. During summer sampling, discrete samples were collected over the water by boat and along the shore downwind of the lake. Flux chamber measurements were taken near the lakeshore. No walking transects were taken at this site. Field of view is approximately 250 m wide.

**Figure 2: Site Pictures. Pingo 1, Pingo 2, Wetland 2 and Wetland 3 were identified as hotspots of methane production during aerial surveys by Kohnert et al. (2017). Pingo 3, Wetland 1 and Lake 1 were identified by holes in the ice forming on water bodies caused by high rate $CH_4$ ebullition. Channel Seep was identified by anomalously high $CH_4$ ebullition spotted from a helicopter during the summer. Pingo 1, Pingo 3, Wetland 1 and Lake 1 were sampled twice each, once during summer and once during the fall.**


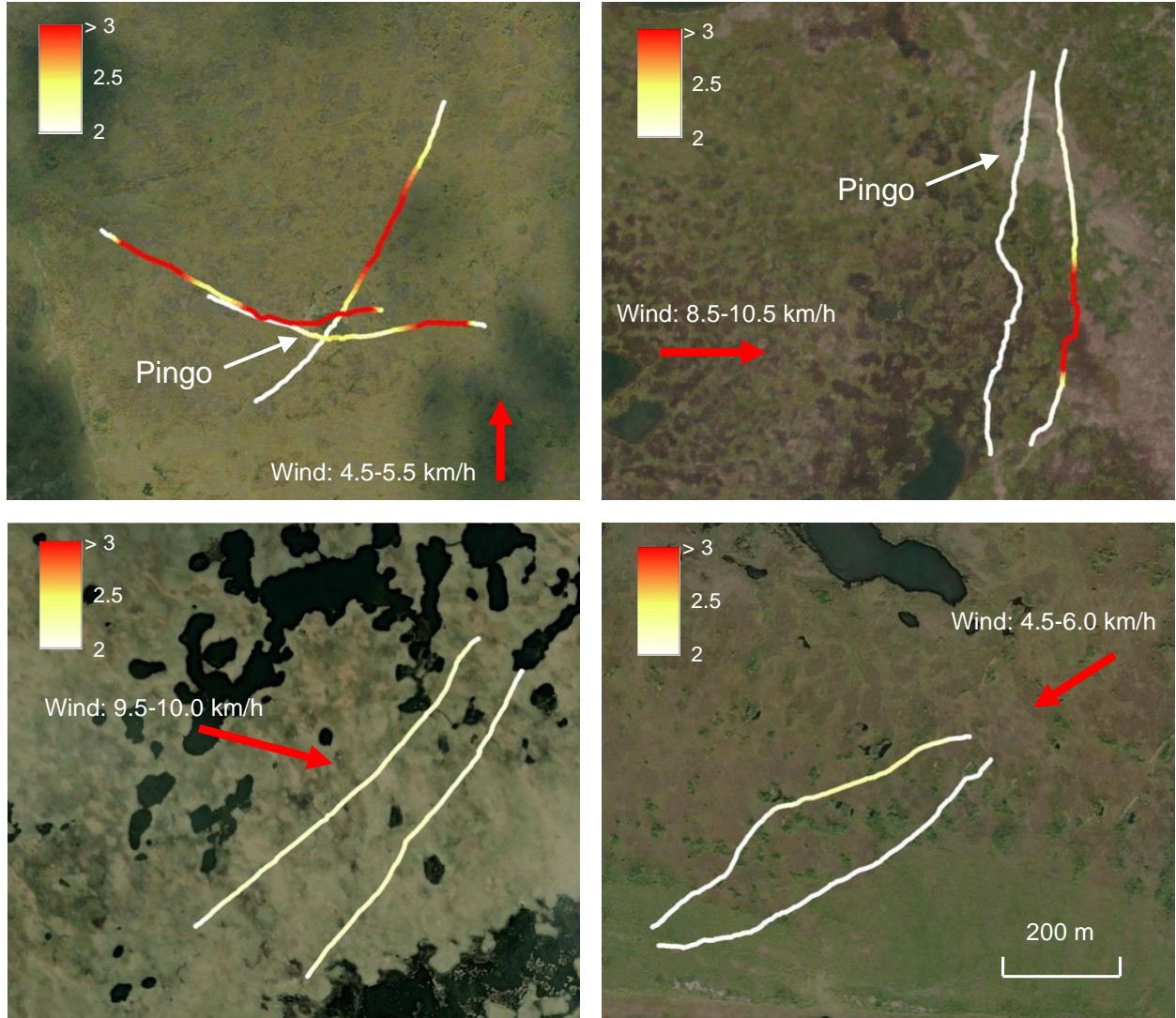

**Figure 3: Walking transects of CH$_4$ concentrations at sample sites, scale is in ppm. The red vector arrow indicates the constant wind direction during the sampling period. Sites are (clockwise from top left): Pingo 1, Pingo 2, Wetland 2, and Wetland 3 (© Google Earth).**

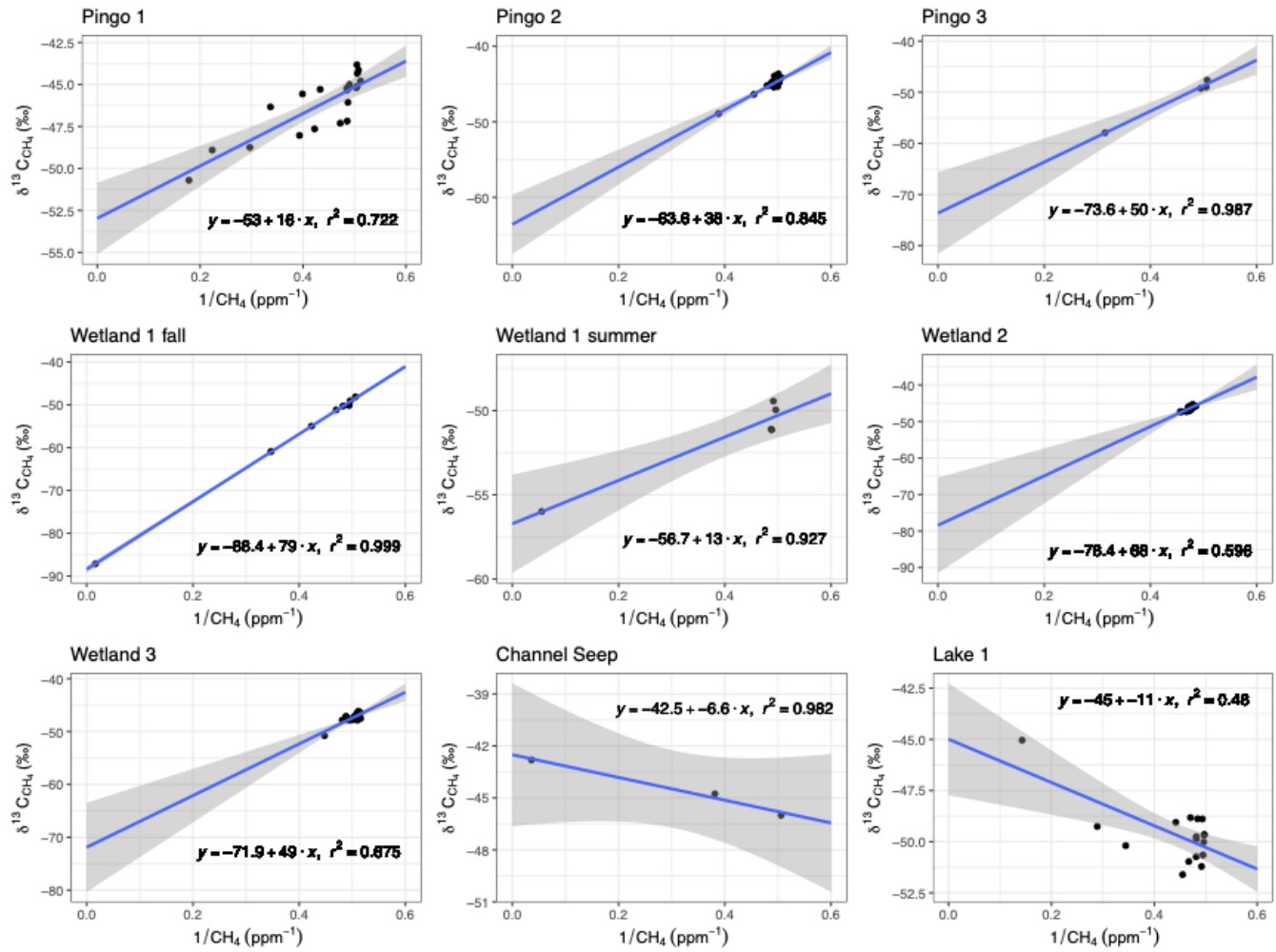


Figure 4: Keeling plots for Pingo 2, Wetland 3, Pingo 1, Wetland 2, Channel Seep, Lake 1, Wetland 1 and Pingo 3. Source signatures ranged from -42 to -88 ‰ $\delta^{13}CH_4$, which includes both thermogenic to biogenic signatures. Channel Seep and Lake 1 had thermogenic signatures, Pingo 3, Wetland 1, Wetland 2 and Wetland 3 had biogenic signatures while Pingo 1 and Pingo 2 had signatures which could be produced from oxidation of biogenic $CH_4$. The grey region indicates the 95% confidence interval of the
regression line.

**Table 1: Summary statistics of CH$_4$ and CO$_2$ mixing ratios determined from walking transects. Elevated CH$_4$ was measured at all sites but concentrations at Pingo 1, Pingo 2, and Wetland 1 were much higher than the other three. Elevated CO$_2$ was observed at Wetland 2, Pingo 2, Pingo 3 and Wetland 1.**

| CH$_4$ (ppm) | | | | | | |
|---|---|---|---|---|---|---|
| **Site** | **Pingo 1** | **Pingo 2** | **Pingo 3** | **Wetland 1** | **Wetland 2** | **Wetland 3** |
| mean | 3.047 | 2.479 | 2.045 | 2.596 | 2.093 | 1.980 |
| median | 2.655 | 1.972 | 2.040 | 2.256 | 2.109 | 1.950 |
| min | 1.971 | 1.676 | 1.974 | 2.061 | 1.628 | 1.707 |
| max | 6.135 | 8.734 | 2.946 | 12.399 | 2.152 | 2.264 |
| sd | 0.955 | 1.479 | 0.039 | 1.049 | 0.071 | 0.107 |

| CO$_2$ (ppm) | | | | | | |
|---|---|---|---|---|---|---|
| **Site** | **Pingo 1** | **Pingo 2** | **Pingo 3** | **Wetland 1** | **Wetland 2** | **Wetland 3** |
| mean | 389 | 402 | 416 | 415 | 430 | 391 |
| median | 389 | 393 | 413 | 413 | 413 | 393 |
| min | 380 | 359 | 380 | 406 | 342 | 361 |
| max | 396 | 462 | 519 | 488 | 603 | 400 |
| sd | 5 | 21 | 9 | 8 | 43 | 7 |
| wind speed (km/h) | 5.5 | 10.5 | 3.6 | 5.0 | 10.0 | 6.0 |