# Peer review of "Characterization of atmospheric methane release in the outer Mackenzie River Delta from biogenic and thermogenic sources – Supplemental Material."

_EGUsphere, 2022_

## Author Comment (AC1)

Commentary reply:

Dear Dr. Lapham,

Thank you for volunteering your time to provide these thorough comments on this manuscript.  In response to your main comment the hypothesis statement can be changed to state:

"We hypothesise that the largest hotspots in the MRD include contributions of biogenic and potentially thermogenic $CH_4$ due to the complex nature of potential geologic $CH_4$ sources at depth in the region and the abundance of environmental settings where modern methane is being produced."

Please see below for our responses to individual comments.

Summary: The goal of the paper was to take a first look at the source of the hot spots previously found by aerial surveys using stable carbon isotopes within the Mackenzie River Delta (MRD). The authors state that the MRD has thermogenic gas seepage and biogenic methane atmospheric sources. Given there are so many lakes, this is not surprising, but there should be references for that statement. The authors hypothesize (line 84) that the hot spots come from "thermogenic, biogenic, and mixed sources....". This hypothesis could be strengthened by picking one source based on previous literature. For example, the Kohnert paper clearly suggests that the hot spots are thermogenic because they are so high, but didn't have any detailed source information to support this. This paper does. Or maybe the fact that there are so many lakes emitting biogenic methane means the atmospheric methane would make one hypothesize these hot spots would be biogenic? The strength of this paper is in using stable carbon isotope ratios of methane in air from ground surveys. The paper could benefit from some more details on the sampling sites, the sampling protocols, the assumptions going into the keeling plots, and overall conclusions drawn from the study. Overall, I think the data is interesting, and definitely novel and worthy of publishing. Hopefully the comments below could help streamline the message:

1. The introduction could be streamlined and strengthened. Currently, the text suggests that the MRD has biogenic methane sources, but I couldn't see any references to support that. Maybe work from Lance Lesack's group would be helpful here for in and around Inuvik. On line 60, there is also mention of production of methane in the organic rich active layer but has no reference. There was a recent paper to conduct permafrost incubation studies (Lapham et al., 2021) in Tuk, but that didn't take place in the active layer. In terms of streamlining, the sentence from line 67-68 could be cut; while it is important to measure emissions of methane, since the current study didn't do this, it's not necessary.

Reply: The sentence from line 67-68 can be removed.

The following sentence can be added to the opening paragraph: "Lakes, especially, in the MRD have been shown to be sources of biogenic $CH_4$ (Cunada et al., 2021; McIntosh Marcek et al., 2021).

The first sentence in the fifth paragraph in the introduction can be changed to read: Migration of $CH_4$ through discontinuities in the permafrost is common in regions with thin permafrost similar to the MRD as well as production in the organic rich active layer during anoxic conditions (Barbier et al., 2012; Liebner & Wagner, 2007; Lupascu et al., 2012).

2. The study location and methods sections could be reorganized to be sure the proper information is conveyed in each section. For example, the setting was described in section2, and then there was a "study location" in section 3. The methods section should only give the methods used, and results should be reported in the results section.

Reply: The study location section can be moved to section 2 with the exception of the following sentence which should be moved to the Results section:

"Of the five airborne eddy covariance hotspots investigated, atmospheric $CH_4$ concentrations were significantly above background at four sites (Pingo 1, Pingo 2, Wetland 2, Wetland 3). Only background atmospheric concentrations of $CH_4$ were observed at Site 9 (Table S1), therefore, it was not included in the main analysis."

3. Overall figures are sufficient with some revision. For example, figure 1 could be more informative if the walking transects were shown on the pictures, and the direction of wind and location of seeps were also shown. As it is now, without any labels, it's difficult to see where the seeps are (unless you know), and at Lake 1, unless you know the picture shows the lake is ice covered, it's hard to know what is happening. The caption says "prominent ebullition" but unless you know what to look for, it's not clear with the captions. Furthermore, supplemental figure S2 is not mentioned in the text, yet addresses some concerns about wind direction and the location of the transects. If the quality could be improved, or information combined between figure 1 and figure S2, this could help.

Reply: It is our opinion that adding the walking transect data to Figure 1 would be too much information to effectively convey in one Figure. But in response to this comment Figure S2 can be moved to the main results section of the paper and the images improved. The remaining two walking transects can be added as well.

4. The chamber flux data seems to be an add-on. Is it necessary for the message of the paper? If so, more details will be needed (like how the samples were analyzed) and some context of what the fluxes mean in terms of other environments (for the discussion). If not needed, please consider taking out of the paper.

Reply: Although the chamber flux data do not add significantly to the narrative presented in this manuscript we had originally included them because they have considerable value. There is a lack of published chamber measurements in the Mackenzie River Delta as well as the Canadian Arctic. The chamber flux data will be removed from the manuscript.

Detailed comments:

Line 34: add in "oxidation" after production and before transport. I think it's important with some of the conclusions drawn to get the idea of oxidation into the text earlier.

Reply: "Oxidation" can be added to line 34

Line 51: As numbers, delta values can be high or low, positive or negative, but not heavy or light. Please use "high" or "low" $\delta^{13}C$ values. Be sure to check this throughout and change accordingly.

Reply: $\delta^{13}C$ values can be referred to as "high" or "low" throughout the manuscript

Line 75: "geologic origin": maybe define what you mean and that it could be made up of thermogenic methane produced deep and migrates, or biogenic or rather, microbial methane produced in the permafrost? See comment from line 224.

Reply: The following can be added to line 75: "($CH_4$ produced beneath the permafrost, including thermogenic $CH_4$ from natural gas reserves that has the potential to migrate through discontinuities in the permafrost)"

Line 78-80: The sentence starting "interestingly, ...." Is a confusing sentence. How can the sources behave differently than the current understanding? Please reword to make more clear.

Reply: This sentence could be changed to read: "Importantly, isotopic signatures have not been extensively used to determine the source of atmospheric $CH_4$ at hotspots in the outer MRD. Source contributions to atmospheric $CH_4$ could be different than the current understanding of the region and other, similar Arctic environments."

Line 117: it would be helpful to have a mark on the map for Tuk.

Reply: The current map in Figure 1 does not encompass Tuktoyaktuk. The field site located in the Tuktoyaktuk coastlands (Pingo 3) is at the southernmost edge of the ecoregion and about 100 km from Tuk.

Lines around 119: describe your sites a little more than just a name. What is "site 9"? There isn't a description in figure 2, and on the figure 1 map, it looks like it's upwind of the hot spot from Kohnert. What were the winds like when you sampled it? Also, the sentence here of "Of the five airborne eddy covariance hotspots....." is a result. It should be moved to that section. In thinking more about the study design, if the idea was to ground truth the Kohnert hot spots, there is a missed opportunity to bring out some novelty of this study. For example, if we look at the "pingo 1" site, the fact that there is a pingo there is important, right? Does that already add information not gained from the Kohnert study? That you observed a pingo there? And what would that mean, what would the observation of a pingo mean for methane emissions? Aren't they by definition conduits of some sort? Or what about pingos being surrounded by wetlands? Additionally, at wetland 3, the fact that there are wetlands characterizing that hot spot is interesting information. I'd almost envision numbering the hotspots sequentially and then giving them their ground feature names as done on figure 1 (this is just a suggestion and maybe not helpful, it just seems interesting what ground features underlay the aerial hot spots). Yet, such an approach really send home the message that is directly inline with your goal, to groundtruth the aerial survey hot spots.

Reply: The following paragraph demonstrating the importance of pingos to $CH_4$ production can be moved from the discussion to the introduction which will help set up the study design:

"Hodson et al. (2020) found that six pingos in Svalbard had a range of annual flux rates between 76.4 and 364 kg $CH_4$/year and concluded that pingos require further study due to their potential contributions of $CH_4$ to the atmosphere. The outer MRD and the Pleistocene deposits of the Tuktoyaktuk Peninsula are home to about 2363 pingos (Wolfe et al., 2021). Release of $CH_4$ from pingos in the region could represent a significant, unaccounted source of $CH_4$ to the atmosphere making it a critical area for further study of pingos as a source of $CH_4$. Migration of $CH_4$ to the atmosphere from pingos is still poorly understood and additional studies of $CH_4$ production from pingos will help to improve Arctic $CH_4$ emission estimates."

- The sentence "Of the five airborne eddy covariance hotspots....." can be moved to the results section.

Table S1: The raw data should be available somewhere for review. Will it be available in a database somewhere, or as a supplemental table here? Also, in this table, for the "source d $\delta^{13}$C-$CH_4$ values, $R^2$, and max $CH_4$" in the fourth column, those are the same values as in figure 1. I'm not sure what this table is adding except to give exact locations. Can you replace this with the raw data, from which you derive the y-intercept from keeling plots? Also, Table S1 gives a "site type" as Polar V. How is the Polar V a site type? Also, you give into in this table for the low $R^2$ values, what made you pick the $R^2$ value cut offs you did? For

example, you kept an $R^2$ of 0.48 but didn't talk about the site with $R^2$ of 0.434. This sort of thing should be mentioned in the methods.

Reply: This table also serves to show the sites in which we were less confident of a source of $CH_4$. The following explanation of how the sites were chosen can be added to the supplement:

"An outline of the sites sampled during this study are included in Table S1. Pingo 1, Lake 1, Wetland 2 and Wetland 3 were included in the main analysis despite low $R^2$ values. The Keeling plot intercept for Wetland 2 and Wetland 3 were low enough that even with a lower confidence in the signature we can still have confidence that it is within the range of biogenic values. Pingo 1 and Lake 1 were included because high $CH_4$ concentration samples clearly indicated a shift towards a thermogenic source despite the poor fit of the keeling plot. This adds confidence to the interpretation even with a low $R^2$ value. A few high concentration samples cannot verify the source signature it can indicate that the source has a signature that is higher (thermogenic) or lower (biogenic) that the atmospheric background value."

The site type "Polar V" can be changed to "Airborne EC"

Line 123: how did you determine where discrete point samples were collected? What was the strategy? Upwind, downwind, etc? This is a study location section, is this the best place for that information? I think you should be explicit that the strategy was to target the hot spots, and if you adopt a sample numbering scheme like numbering the hot spots, this strategy will be clear.

Reply: The following can be added to the end of the first paragraph in the sample collection section of the methods (3.2, line 165):

"Walking transect locations were selected by completing one transect up wind and one down wind of the estimated location of the highest flux concentration, in order to obtain background concentrations at each site. Discrete point samples were taken parallel to each walking transect, 3-5 m further away from the center of the hotspot. This sampling method was designed to cover the most area, with the fewest samples, in the shortest amount of time possible. By using this method we were still only able to cover a small fraction of the hotspot (5-10% by area), so samples were collected around potential sources of $CH_4$, such as pingos or wetlands. This increased the likelihood of pinpointing the source."

Line 125: Is "Lake 1" known as another lake by the community? Is that "shot hole" lake or Swiss cheese? And Channel Seep, is that channel seep 1? There could be reports that have some isotope data reported to help aid you in interpretations.

Reply: Lake 1 is locally known as Swiss Cheese Lake. The Geological Survey of Canada (GSC) has previously sampled Channel Seep but the results were not published. The results of the previous samples collected by the GSC can be added to section 5.2 of the discussion.

Line 126: the "observations of ebullitions seen in open water in summer", were those your observations made in this sampling trip or previous knowledge? Cite previous knowledge.

Reply: The following can be added to the second paragraph of the study location section: "Channel Seep and Lake 1 were previously known to researchers in the field party, holes in the ice were observed from the helicopter while passing over Pingo 3 and Wetland 1 sites during the fall."

Line 126: how close is "as close as possible" to observed ebullitions? Were you on land, upwind, downwind? How long were walking transects, 10m? This discussion in these 3 lines around 130 are more about sample collection. As such, they should be moved to that section of the paper.

Reply: The following can be added to the second paragraph in the sample collection section:

"Point samples at aquatic seep sites were taken as close as possible to observed ebullitions with additional samples taken up to 5 m away from known sources in order to measure source and background $CH_4$ concentrations. At Pingo 3 and Wetland 1 point samples were taken directly over holes in the ice by throwing a buoy with tubing attached to it onto the water in the open hole in the ice and using a pump to draw air through the tubing into a sample bag. At Lake 1 samples were collected from a boat and positioning the sample inlet directly above seeps. Channel Seep samples were collected by extending a pole with the sample inlet attached to the end out over the seep from the channel bank. All samples were taken from the air less than 10 cm from the seep with the exception of the channel seep sample which was approximately 2 m from the seep. Walking transects and discrete point samples were completed at Pingo 3 and Wetland 1 in the summer of 2021 to compare overall site variation to point source samples."

Lines 131-142: This paragraph seems more high level than where it is situated in the paper. I suggest reorganizing this study location section.

Reply: The Study Location section can be combined with the setting section.

Line 149: Are you missing the word "under"? Permafrost "under" Tuk.....

Reply: This can be edited to read "The permafrost of the Tuktoyaktuk Coastlands..."

Line 154: what is precision of handheld GPS and what model, make? What is the precision of the licor and the picarro CRDS? It's important to mention the cavity ring down spectrometer (CRDS) part of the instrument since that is how the measurement is made. And what standards were used to calibrate the concentration and isotope measurements? The placement of the analysis instruments seems out of place here since this is the sample collection section.

Reply: The section can be renamed "Sample Collection and Analysis

beginning of the section can be rewritten as follows:

"Geolocated air samples were collected in the field and subsequently analysed at the Aurora Research Institute laboratory in Inuvik, Northweset Territories, Canada or at St. Francis Xavier University in Antigonish, Nova Scotia, Canada. $CH_4$ and $CO_2$ concentrations and $\delta^{13}C$-$CH_4$ were determined using a Picarro G2210i analyser. The Picarro G2210i analyser is accurate to within 1 ‰ $\delta^{13}C$-$CH_4$, 0.001 ppm $CH_4$, and 1 ppm $CO_2$. All point samples were analyzed for at least 5 minutes and the values averaged, which increases the Picarro G2210i analyser accuracy to 0.1 ‰ $\delta^{13}C$-$CH_4$, 0.0001 ppm $CH_4$ and 0.01 ppm $CO_2$. Only point samples were used for the determination of $\delta^{13}C$-$CH_4$ source signatures. Sample positions were recorded with a Garmin eTrex 10 handheld GPS, which has an accuracy of +/- 3.65 m. Gas mixing ratios were standardized to Ameriflux FB04306 breathing grade air (benchmarked to 0.5 ppb)"

The following can also be added:

"The Li-Cor LI-7810 gas analyser has an accuracy of 0.25 ppb."

Line 156: Please cite the airborne work paper, since it was not done in this study.

Reply: Kohnert, 2017 will be cited here.

Line 158: "photographs of each site…." I think this should be moved to study location, and not sample collection. You are using the photographs to describe the sites, correct? If so, they really set the stage for the setting, which belongs elsewhere.

Reply: This sentence can be moved to the end of the second paragraph in the study location section.

Line 159: "walking transections…" what was pumping rate? What are dimensions of tubing? Is 6mm OD or ID?

Reply: The pumping rate was 20 standard cubic centimeters per minute. 6 mm was the outside diameter, it can be changed to the inside diameter of 4 mm in the text which is more relevant. This sentence can be changed to read as follows: " Walking transects were carried out by filling a 30 m coil of 4 mm inside diameter aluminium Synflex tubing at a constant rate of 20 standard cubic centimeters per minute while walking at a steady pace across the ground or by carrying a Li-Cor LI-7810 gas analyser."

Line 163: "Mixing between sample collection and analysis is limited due to small diameter of tubing". Have you proved that? Or is there a paper you can cite for this? The reason I ask is that the pumping rate is pretty fast, so I would imagine your sample will smear alone the edges of the tubing and mix along the way it's filling. Please give more details as to the accuracy of this approach.

Reply: The method used in this manuscript had been adapted from a study referenced on line 164 (Andersen et al., 2018). The original method used the same type and diameter tubing mounted on a drone and filled it at a similar flow rate (21.5 sccm). The main difference between the two methods is that the original study used an orifice to control the flow while this study used a flow controller. In addition, the authors of the current study tested the method in the lab by filling the lengths of tubing used during the fieldwork with concentrations of $CH_4$ interspersed with Nitrogen in concentrations between 2.5 ppm up to 10 ppm $CH_4$. The results showed that there was no distortion on filling the tubing but substantial mixing occurred after 24 hours. In order to prevent mixing during the fieldwork samples were analyzed immediately on return from the field site (within 8 hrs). A description of this experiment can be added to the supplemental methods.

Line 166: Why did you pick 1 meter above ground level? Did you ever try to go down to ground level? Did you see any change in the concentration? Or is 1m desirable because things are more mixed and you are trying not to see a ground signal? What is the thought behind this?

Reply: Because $CH_4$ source locations were unknown we sampled air at 1 m so that was adequately mixed but still practical to sample. Sampling closer to the ground would have allowed us to sample closer to potential sources but would have reduced our ability to sample sources that were further away from our sample location.

Line 171: Where are the flux chamber measurement data? Also, this section is confusing as written. It looks like 2 chambers were used, the automated one from licor, but I can't tell what was used for Lake 1, using manual extraction of samples. And the dimensions are for a flux chamber, but there is only one, yet it seems two chambers were used? Also, can you discuss how only allowing 1 hour between collars installed and measurement might change your results?

Reply: The soil disturbance caused by collar installation can create a flush of $CO_2$ (Bahn et al., 2010), therefore, $CO_2$ fluxes measured during this study may be slightly high because we were unable to wait the recommended 24 hours.

The flux chamber data and corresponding methods can be removed from the paper. Please see main comment 4 above.

Line 178: "Keeling plot analysis". I am not sure this is the right terminology. You analyzed keeling plots to determine the stable carbon isotope signature of the methane source. That phrase, keeling plot analysis, is used several times, so maybe it is the right term but it could be better to say "We constructed keeling plots with the discrete transect data to determine …." And also cite the "common approach" of using keeling plots. I agree that it is common, but I am not sure it is common to take discrete measurements in the horizontal direction versus the vertical direction. Meaning, I thought keeling plots were always done in the vertical collection of air in a forest canopy, for example. If that is not true, it's probably still making a note of this since it does seem a bit novel to use this technique for the walking transects. Or maybe there are papers to show this approach used in this way.

Reply:  While the use of keeling plots is common, they are used with a variety of sampling techniques to identify temporal and spatial shifts in $^{13}C$ for $CH_4$ and $CO_2$. It may be more applicable here to say that they are broadly used and give some examples of similar uses. While this particular method may not be common, it would be a stretch to call it novel.

The first paragraph in section 3.3 could be changed to read:

"Keeling plots were constructed from the discrete point samples and used to determine the stable carbon isotope signature of the $CH_4$ source at each site. Keeling plots are broadly used to determine a source of carbon entering the atmosphere by measuring the change in $\delta^{13}C$-$CH_4$, or fractionation, that occurs as more carbon from that source is added (Kohler et al., 2006). The Keeling plot method has previously been used to measure spatial variation of $\delta^{13}C$-$CH_4$ in anthropogenic (Chamberlain et al., 2016; Zazzeri et al., 2015, 2017) and natural sources (Skorokhod et al., 2016)."

Line 193: Do you have the bubble isotope values from historical data by the GSC to compare to your point measurements?

Reply: Yes, we have access to unpublished data from the GSC for lake 1. The following can be added to section 5.2 in the discussion:

"Analyses of a seep gas samples from this lake carried out in 2008 yielded values of -290 ‰ $\delta D$-$CH_4$/ -45 ‰ $\delta^{13}C$-$CH_4$ and -230 ‰ $\delta D$-$CH_4$/ -37‰ $\delta^{13}C$-$CH_4$ (S.R. Dallimore, personal communication, January 12th, 2023) This is well within the thermogenic isotopic field as determined by Whiticar (1999)."

Line 197: Why is walking transect data shown as average data? The maximum value of 12ppm is very interesting and seems important to know where you were in comparison to the "wetland". Is it possible to do this from the GPS location data you obtained?

Reply: Yes, the raw data for the walking transects could be added to the supplement

Line 203: The sentence of "estimates of source…." Seems a bit premature. Since this is the methods section, could you first say that the keeling plots are shown in figure 3, and show intercepts of X, Y, Z, which indicate the source of the methane? And also give the R2 values? As it is now, you don't mention figure 3, so it's unclear where these numbers come from. The figure 3 caption mentions a "grey region" but there are no grey regions on the figures. And finally, there is a formatting issue with the equation written on the figures. It is also important to mention the cut off you used for the R2 values.

Reply: The sentence on line 203 could be changed to read: "Source stable carbon isotope signatures ($\delta^{13}$C-CH$_4$) were derived from Keeling plot Y intercept values (Figure 3) and were 53.0 ‰ for Pingo 1, -63.6 ‰ for Pingo 2, -78.4‰ for Wetland 2, and -71.9 ‰ for Wetland 3."

The plots will be updated with grey regions and the extra 'x' can be removed from the figure.

Line 208: "Keeling plot values" is not quite accurate. Maybe say "keeling plot y-intercepts". It's interesting the seasonal component of these values. Is seems reasonable to think that during the winter, there isn't as much methane oxidation, which leads to the lower values. I think you mention that in discussion.

Reply:  Keeling plot values can be changed to "keeling plot y-intercepts."

Line 211: "Flux rates" isn't accurate. A flux is a concentration per area per time, a rate is distance per time. I think you just mean fluxes here. It also seems like the chamber fluxes are an afterthought since the fluxes aren't given here. And they are put in a supplemental table. Are they needed?

Reply: Flux rates can be changed to "fluxes". Please see main comment 4 above.

Lines 215-218: change "values" to "concentrations".

Reply: values can be changed to "concentrations"

Line 217: take out "were". What is an "observation" in this context? There wasn't mention of observations before now. Where does the 2013 number come from? Do you mean the discrete measurements? Same question for the 1850 observations at pingo 1. This section is a bit confusing since it's also written like results, but yet in the discussion section.

Reply:

The following sentence can be moved to the results section: "During the walking transects concentrations above 2.5 ppm were recorded for 1056 of 1850 measurements at Pingo 1 and for 285 of 2013 measurements at Pingo 2."

The paragraph starting on line 215 can be rewritten to read:

"The highest $CH_4$ concentrations for walking transects co-located within airborne eddy covariance $CH_4$ hotspots were obtained in the north-western part of the outer MRD. The elevated values measured over a dispersed area within the eddy covariance hotspots provides a basis to speculate on the possible sources for the hotspots. Estimates of source stable carbon isotope signatures ($\delta^{13}C$-$CH_4$) derived from Keeling plots were -53.0 ‰ for Pingo 1 and -63.6 ‰ for Pingo 2 with reasonable confidence in these determinations as they were derived with multiple isotopic measurements for each Keeling plot. These signatures are substantially enriched in $^{13}C$ compared to typical biogenic sources and relatively close to our assumed threshold of -50 ‰ based on thermogenic gas found in the Taglu hydrocarbon reservoir. We conclude, therefore, a geologic source for the methane for each site is made up of a mixture of thermogenic and biogenic gas from depth, perhaps with a dominance of thermogenic methane. However, while elevated values for the walking transects were observed in close proximity to the pingo features, we note that for both sites the highest values were not on the features themselves, but a short distance away in the low lying shrub tundra terrain surrounding the pingos."

Line 224: For the general reader, it might help to define geologic source in the introduction to put this conclusion in more context.

Reply: The following phrase can be added to line 26 of the introduction to define Thermogenic as used in this paper: "The area is characterized by thin and destabilizing permafrost (Burn & Kokelj, 2009), high organic content soils (Schuur et al., 2008), vast amounts of deep thermogenic methane, originating from fossil hydrocarbon reservoirs (Collett & Dallimore, 1999) and over 49,000 lakes...".

This is the same definition used by Walter Anthony et al. (2012) .

Line 240: Can you give values for the pingo 1 and 2 site in the text? It would help the reader not have to flip back to the figures.

Reply: The values of -53.0 ‰ for Pingo 1 and -63.6 ‰ for Pingo 2 can be added to the text here.

Line 241: Is this new data? I didn't see the reporting of the methane concentrations over the pingo features themselves. And this sentence is also a bit confusing as to what you

mean. Did you do 2 transects for Pingo 1? Seems like supplemental figure S2 (top right) could be helpful to show what you mean here. You could refer to that figure here, but that figure quality needs to be improved.

Reply: The sentence on line 241 can be rewritten to read: "A point of interest, however, is that the highest $CH_4$ concentrations at sites Pingo 1 and Pingo 2 were measured 200-400 m (Figure 2) away from the pingos and were only as high as 2.2 ppm directly over the pingos (Table S3)."

Figure S2 can be moved to the results section (now Figure 2) of the main paper and the images improved and the raw data from the walking transects has been added to the supplement (Table S3).

Line 243: Only wind speed is reported in table 1. Wind direction is also key that would be important to show in that table. Or did you always collect samples downwind? Do you see a correlation with wind speed? I would think that the higher the wind speed, the further away the source of that gas could be.

Reply: Samples were collected upwind and downwind of any features so that we would be certain to pick up background and peak concentrations if the feature were a source. Wind speed and direction is shown of Figure S2, which can be moved to the main manuscript. We have refrained from trying to make correlations with wind speed because we do not have high-rate wind measurements. The following sentence could be added to the sample collection section of the methods:

"If features that represented potential sources such as pingos or wetlands were present, than walking transects were taken upwind and downwind of the feature."

Paragraph starting at line 254: This is a great reason you chose to sample pingos. Could you move it to the introduction to help set up the "why" for your study? After reading the paper again, it seems you didn't set out to study pingos, per say, but you found those pingos at the ground features under the aerial hot spots. If you present these ground features as part of the results, I don't think you need to describe why you chose pingos to study, but instead, it will be clear why you sampled them.

Reply: This paragraph could be moved to the third paragraph in the introduction.

Line 262: "these sites had no obvious geologic...." What sites are being referred to here: wetland 1, 2 and 3? If so, is there a hot spot at wetland 1? It's not obvious on the map. And wetland 3 is very close to swiss cheese seeps, correct? Seems like a potential geologic source.

Reply: The sentence on line 262 can be removed.

There is a typo in line 261, three should be two, referring to Wetland 2 and Wetland 3 which are discussed in this paragraph. This has been corrected. Wetland 1 was not identified by the arial eddy covariance study, therefore it does not show up on the map in Figure 1.

Wetland 3 is indeed close (2.5 km) to Swiss Cheese Lake (Lake 1 in this study). This does show that there is potential for geologic sources in the area but not necessarily at the site. Permafrost conditions, for example, could be very different than Swiss Cheese Lake which is much deeper and more likely to have talik formation underneath which increases the likelihood of geologic sources at that site.

Line 268: Can you add in a sentence after "….Wetland 3."?  Please consider adding in: "Our data is consistent with knowledge that wetlands produce significant methane from microbial degradation. This carbon is also probably recent in age versus geologic methane." And you can give some citations for that.

Reply: The following sentence can be added:  "This is consistent with knowledge that wetlands can produce significant amounts of biogenic $CH_4$ (Andresen et al., 2017; McGuire et al., 2012; Wik et al., 2016)"

Line 269: For the lack of signal at site 9, were you downwind of the wetland?

Reply: Yes, the wind was very light, below 3.5 km/h and we sampled upwind and downwind of the wetland. The site was actually quite dry and the wetland had been reduced to a few metres across which would likely have reduced the emissions. Older satellite image shows a clearly larger wetland so the wetland was likely deeper in the past.

Line 273: Can you give reference for this?

Reply:  Chanton et al. (2005) can be cited here.

Line 288: What is the evidence for thermogenic methane at this site? Is there any reason to think that the seep could be thermogenic? -53 from the keeling intercept still seems quite low for thermogenic. It just seems that this is most likely an oxidation signal. But as you say, it is still possible there could be thermogenic.

Reply: The authors may have understated the possibility of oxidation at this site during the summer to create the signature of – 53‰.  The paragraph can be reworded as follows to better illustrate the two possibilities at this site:

Discrete sampling at Wetland 1 yielded  $\delta^{13}$C-$CH_4$ Keeling plot source signature of -88.3 ‰ when sampled in October during freeze up and -53.4 ‰ during the summer. In simple terms, this suggests a biogenic source when sampled in October but a more complex scenario during the summer. While the sampling was carried out at the same location,

methane ebullition was seen while sampling during the fall, but not during the summer. Biogenic production can persist late into the cold season (Zona et al., 2016) so it is not surprising to see high-rate biogenic production during freeze up. The lack of ebullition flux at the same site during the summer and the different Keeling plot estimate suggest methane flux in this wetland setting varies seasonally. The Keeling plot source signature of -53.4 ‰ during the summer could be caused from either oxidation of a biogenic source or contributions of both biogenic and thermogenic sources. Seasonal shifts in lake-produced $CH_4$ stable carbon isotope signatures potentially due to oxidation are known to occur but are typically observed during winter beneath ice cover (Ettwig et al., 2016; Michmerhuizen et al., 1996), or the transition from the ice covered to open water periods in the spring (McIntosh Marcek et al., 2021). Similar observations for seasonal variability in terrestrial sources are not well documented in the literature, although the Wetland 1 site was dominated by sedge vegetation with areas of standing water and transport of $CH_4$ from anaerobic soils with sedge vegetation has been observed to bypass the aerobic zone, limiting oxidation during the growing season (King et al., 1998; Olefeldt et al., 2013). Therefore, it is possible that there were contributions to the atmosphere from biogenic and thermogenic sources at Wetland 1, but oxidation of biogenic $CH_4$ during the summer cannot be ruled out as the reason for the signature derived during the summer sampling.

Line 294: I believe the location of Lake 1 is the same as "swiss cheese" that has been visited by the GSC before. Are there reports that report the bubble signature isotope value? It might be informative in your discussion of your values.

Reply: Yes, it is the same location. We have access to data from the GSC from seep gas analysis at the lake that is not published. The following sentence can be added here.

"Analyses of a seep gas sample from this lake carried out in 2008 yielded values of -290 ‰ $\delta D$-$CH_4$/ -45 ‰ $\delta^{13}C$-$CH_4$ and -230 ‰ $\delta D$-$CH_4$/ -37‰ $\delta^{13}C$-$CH_4$ (S.R. Dallimore, personal communication, January 12th, 2023) This is well within the thermogenic isotopic field as determined by Whiticar (1999)."

**Cited Literature:**

Andresen, C. G., Lara, M. J., Tweedie, C. E., & Lougheed, V. L. (2017). Rising plant-mediated methane emissions from arctic wetlands. *Global Change Biology*, *23*(3), Article 3. https://doi.org/10.1111/gcb.13469

Bahn, M., Kutsch, W. L., Heinemeyer, A., & Janssens, I. A. (2010). Appendix: Towards a standardized protocol for the measurement of soil $CO_2$ efflux. In W. L. Kutsch, M. Bahn, & A. Heinemeyer (Eds.), *Soil Carbon Dynamics* (1st ed., pp. 272–280). Cambridge University Press. https://doi.org/10.1017/CBO9780511711794.016

Barbier, B. A., Dziduch, I., Liebner, S., Ganzert, L., Lantuit, H., Pollard, W., & Wagner, D. (2012). Methane-cycling communities in a permafrost-affected soil on Herschel Island, Western

Canadian Arctic: Active layer profiling of *mcrA* and *pmoA* genes. *FEMS Microbiology Ecology*, *82*(2), 287–302. https://doi.org/10.1111/j.1574-6941.2012.01332.x

Burn, C. R., & Kokelj, S. V. (2009). The environment and permafrost of the Mackenzie Delta area. *Permafrost and Periglacial Processes*, *20*(2), 83–105. https://doi.org/10.1002/ppp.655

Chamberlain, S. D., Ingraffea, A. R., & Sparks, J. P. (2016). Sourcing methane and carbon dioxide emissions from a small city: Influence of natural gas leakage and combustion. *Environmental Pollution*, *218*, 102–110. https://doi.org/10.1016/j.envpol.2016.08.036

Chanton, J., Chaser, L., Glasser, P., & Siegel, D. (2005). Carbon and Hydrogen Isotopic Effects in Microbial, Methane from Terrestrial Environments. In *Stable Isotopes and Biosphere Atmosphere Interactions* (pp. 85–105). Elsevier. https://doi.org/10.1016/B978-012088447-6/50006-4

Collett, T. S., & Dallimore, S. R. (1999). Hydrocarbon gases associated with permafrost in the Mackenzie Delta, Northwest Territories, Canada. *Applied Geochemistry*, *14*(5), 607–620. https://doi.org/10.1016/S0883-2927(98)00087-0

Cunada, C. L., Lesack, L. F. W., & Tank, S. E. (2021). Methane emission dynamics among CO2-absorbing and thermokarst lakes of a great Arctic delta. *Biogeochemistry*, *156*(3), 375–399. https://doi.org/10.1007/s10533-021-00853-0

Ettwig, K. F., Zhu, B., Speth, D., Keltjens, J. T., Jetten, M. S. M., & Kartal, B. (2016). Archaea catalyze iron-dependent anaerobic oxidation of methane. *Proceedings of the National Academy of Sciences*, *113*(45), 12792–12796. https://doi.org/10.1073/pnas.1609534113

King, J. Y., Reeburgh, W. S., & Regli, S. K. (1998). Methane emission and transport by arctic sedges in Alaska: Results of a vegetation removal experiment. *Journal of Geophysical Research: Atmospheres*, *103*(D22), Article D22. https://doi.org/10.1029/98JD00052

Kohler, P., Fischer, H., Schmitt, J., & Munhoven, G. (2006). *On the application and interpretation of Keeling plots in paleo climate research – deciphering $\delta^{13}C$ of atmospheric $CO_2$ measured in ice cores*. 18.

Liebner, S., & Wagner, D. (2007). Abundance, distribution and potential activity of methane oxidizing bacteria in permafrost soils from the Lena Delta, Siberia. *Environmental Microbiology*, *9*(1), 107–117. https://doi.org/10.1111/j.1462-2920.2006.01120.x

Lupascu, M., Wadham, J. L., Hornibrook, E. R. C., & Pancost, R. D. (2012). Temperature Sensitivity of Methane Production in the Permafrost Active Layer at Stordalen, Sweden: A Comparison with Non-permafrost Northern Wetlands. *Arctic, Antarctic, and Alpine Research*, *44*(4), 469–482. https://doi.org/10.1657/1938-4246-44.4.469

McGuire, A. D., Christensen, T. R., Hayes, D., Heroult, A., Euskirchen, E., Kimball, J. S., Koven, C., Lafleur, P., Miller, P. A., Oechel, W., Peylin, P., Williams, M., & Yi, Y. (2012). An assessment of the carbon balance of Arctic tundra: Comparisons among observations, process models, and atmospheric inversions. *Biogeosciences*, *9*(8), 3185–3204. https://doi.org/10.5194/bg-9-3185-2012

McIntosh Marcek, H. A., Lesack, L. F. W., Orcutt, B. N., Wheat, C. G., Dallimore, S. R., Geeves, K., & Lapham, L. L. (2021). Continuous Dynamics of Dissolved Methane Over 2 Years and its Carbon Isotopes ($\delta^{13}C$, $\Delta^{14}C$) in a Small Arctic Lake in the Mackenzie Delta. *Journal of Geophysical Research: Biogeosciences*, *126*(3). https://doi.org/10.1029/2020JG006038

Michmerhuizen, C. M., Striegl, R. G., & McDonald, M. E. (1996). Potential methane emission from north-temperate lakes following ice melt. *Limnology and Oceanography*, *41*(5), 985–991. https://doi.org/10.4319/lo.1996.41.5.0985

Olefeldt, D., Turetsky, M. R., Crill, P. M., & McGuire, A. D. (2013). Environmental and physical controls on northern terrestrial methane emissions across permafrost zones. *Global Change Biology*, *19*(2), Article 2. https://doi.org/10.1111/gcb.12071

Schuur, E. A. G., Bockheim, J., Canadell, J. G., Euskirchen, E., Field, C. B., Goryachkin, S. V., Hagemann, S., Kuhry, P., Lafleur, P. M., Lee, H., Mazhitova, G., Nelson, F. E., Rinke, A., Romanovsky, V. E., Shiklomanov, N., Tarnocai, C., Venevsky, S., Vogel, J. G., & Zimov, S. A. (2008). Vulnerability of Permafrost Carbon to Climate Change: Implications for the Global Carbon Cycle. *BioScience*, *58*(8), 701–714. https://doi.org/10.1641/B580807

Skorokhod, A. I., Pankratova, N. V., Belikov, I. B., Thompson, R. L., Novigatsky, A. N., & Golitsyn, G. S. (2016). Observations of atmospheric methane and its stable isotope ratio (δ13C) over the Russian Arctic seas from ship cruises in the summer and autumn of 2015. *Doklady Earth Sciences*, *470*(2), 1081–1085. https://doi.org/10.1134/S1028334X16100160

Sriskantharajah, S., Fisher, R. E., Lowry, D., Aalto, T., Hatakka, J., Aurela, M., Laurila, T., Lohila, A., Kuitunen, E., & Nisbet, E. G. (2012). Stable carbon isotope signatures of methane from a Finnish subarctic wetland. *Tellus B: Chemical and Physical Meteorology*, *64*(1), 18818. https://doi.org/10.3402/tellusb.v64i0.18818

Walter Anthony, K. M., Anthony, P., Grosse, G., & Chanton, J. (2012). Geologic methane seeps along boundaries of Arctic permafrost thaw and melting glaciers. *Nature Geoscience*, *5*(6), Article 6. https://doi.org/10.1038/ngeo1480

Whiticar, M. J. (1999). Carbon and hydrogen isotope systematics of bacterial formation and oxidation of methane. *Chemical Geology*, *161*(1–3), Article 1–3. https://doi.org/10.1016/S0009-2541(99)00092-3

Wik, M., Varner, R. K., Anthony, K. W., MacIntyre, S., & Bastviken, D. (2016). Climate-sensitive northern lakes and ponds are critical components of methane release. *Nature Geoscience*, *9*(2), 99–105. https://doi.org/10.1038/ngeo2578

Zazzeri, G., Lowry, D., Fisher, R. E., France, J. L., Lanoisellé, M., Grimmond, C. S. B., & Nisbet, E. G. (2017). Evaluating methane inventories by isotopic analysis in the London region. *Scientific Reports*, *7*(1), Article 1. https://doi.org/10.1038/s41598-017-04802-6

Zazzeri, G., Lowry, D., Fisher, R. E., France, J. L., Lanoisellé, M., & Nisbet, E. G. (2015). Plume mapping and isotopic characterisation of anthropogenic methane sources. *Atmospheric Environment*, *110*, 151–162. https://doi.org/10.1016/j.atmosenv.2015.03.029

Zona, D., Gioli, B., Commane, R., Lindaas, J., Wofsy, S. C., Miller, C. E., Dinardo, S. J., Dengel, S., Sweeney, C., Karion, A., Chang, R. Y.-W., Henderson, J. M., Murphy, P. C., Goodrich, J. P., Moreaux, V., Liljedahl, A., Watts, J. D., Kimball, J. S., Lipson, D. A., & Oechel, W. C. (2016). Cold season emissions dominate the Arctic tundra methane budget. *Proceedings of the National Academy of Sciences*, *113*(1), Article 1. https://doi.org/10.1073/pnas.1516017113

---

## Author Comment (AC2)

Anonymous referee # 1 comments and reply.

Dear Referee # 1,

Thank you for your contribution in reviewing this manuscript. We have outlined considerable changes that could be made to the manuscript in order to address your concerns that oxygenation of biogenic methane could produce the $\delta^{13}$C-CH$_4$ signatures observed in this study.

This manuscript aims to determine the contribution of biogenic and thermogenic methane (CH$_4$) to CH$_4$ fluxes from the Mackenzie River Delta (MRD) into the atmosphere. Therefore, the authors collected surface air samples from several sampling sites in the MRD and analysed their CH$_4$ concentration and the $\delta^{13}$C value of CH$_4$. To differentiate between thermogenic and biogenic CH$_4$ they used two thresholds, assuming that CH$_4$ with a carbon stable isotope value of > -50‰ is of thermogenic origin and CH$_4$ with a value < -70‰ of biogenic origin. Values between -50‰ and -70‰ would indicate a mixture of both sources. The main conclusions of the manuscript are based on this assumption, which is, however, a substantial oversimplification. There are numerous studies demonstrating carbon stable isotope signatures of biogenic CH$_4$ of > -70‰, in particular CH$_4$ from acetoclastic methanogenesis (see e.g. Bréas et al. (2001), Chanton & Smith (1993), Conrad (2005)), including from permafrost affected wetlands (Nakagawa et al., 2002). Furthermore, CH$_4$ emitted from highly heterogeneous wetlands as the one studied here are affected by microbial CH$_4$ oxidation, which causes the carbon stable isotope signature of released CH$_4$ to increase, also to values above -50‰. There are many studies about the impact of CH$_4$ oxidation in northern wetlands on CH$_4$ fluxes and the carbon stable isotope signatures of released CH$_4$ (e.g. Happell et al. (1994), Vaughn et al. (2016)), but the effect of CH$_4$ oxidation on carbon stable isotope values of CH$_4$ is mentioned only very briefly. Since the carbon stable isotope values of released methane may vary strongly, e.g. due to different CH$_4$ production pathways, CH$_4$ transport and CH$_4$ oxidation, carbon stable isotope values between -42 ‰ and -88 ‰, as presented in this manuscript, may be explained by biogenic sources alone and are also reported for northern wetlands not affected by fluxes of thermogenic methane. Hence, I do not see that carbon stable isotope values of released methane alone provide robust information to answering the central research question of this manuscript, the contribution of biogenic and thermogenic methane to methane release in the MRD. To give substantial information on this question, further data are needed, e.g. the $\delta$D signatures of CH$_4$, its $^{14}$C age, or the concentration of further hydrocarbons.

Furthermore, methods of gas sampling and analysis and calculation of the source $\delta^{13}$C value should be described in more detail. What was the gas flow while flushing the Synflex tube, how often was it flushed with the air sample to ensure that no contaminations remained? How was gas collected with the LI-7810, how often and at which positions? Why were gas samples collected in the Synflex tube, if they were not analysed for d$^{13}$C of CH$_4$? How far is 'as close as possible'? Please clearly describe which samples were collected for which analysis. Particularly for the Keeling-plots and the calculation of the $\delta^{13}$C source values it should be clearly explained from which collected sample the CH$_4$ concentrations and $\delta^{13}$C values were analysed.

Finally I suggest restructuring the Results and Discussion section. In the current version of the manuscript a substantial part of the results are presented (or repeated) in the Discussion. Specific comments:

We agree that measuring deuterium as well as $^{13}CH_4$ would significantly strengthen the conclusions made in this study. We also agree that our interpretation of signatures may be an oversimplification. Sites with stable carbon isotope signatures observed in this study which we have described as "mixed" sources could have potentially been produced by oxidized biogenic methane. Specifically, sites Pingo 1 (-53.0 ‰) and Pingo 2  (-63.6 ‰). We could make changes to the first paragraph of the discussion around our determination of the source of $CH_4$ at these sites. The potential effect of oxidation is somewhat understated in the manuscript. In response, we can add existing literature values for biogenic $CH_4$ with the highest stable carbon isotope signatures to the discussion. We find that that biogenically derived, stable carbon isotope signatures as high as - 50‰ are exceedingly rare for arctic permafrost environments, but they do exist. Values as high as -42 ‰ are not present in the literature. The highest signatures reported in literature for arctic lakes or wetlands are -58.2 in Nakagawa et al. (2002), -52.3 in Vaughn et al. (2016), -49.2 ‰, (Thompson et al., 2016) and -44.9 ‰ (Preuss et al., 2013). The isotopic values from Siberian alasses reported in Nakagawa (2002) were misrepresented in Conrad (2005) as -43 ‰ to -27 ‰ when they are actually -63.9 ‰ to -58.2 ‰. This is the only Arctic site referenced by Conrad (2005). Happell et al. (1994), Chanton & Smith, (1993) and Bréas et al. (2001) observed $^{13}CH_4$ values greater than -50‰ (as high as -37 ‰, -41 ‰, and -31 ‰).  All of these studies were south of 30°N where $\delta^{13}C$ signatures are higher than those found in the Arctic due to the prevalence of C4 plants (Chanton & Smith, 1993; Fisher et al., 2017; Nakagawa et al., 2002; Oh et al., 2022).

Please see below for our responses to individual comments.

L11: To my understanding, $CH_4$ is released but not produced from thermogenic sources. Please clarify

Reply: "Production" can be changed to "emission"

L30: What means 'conductive for biogenic $CH_4$ production'? Please clarify
Reply: "Conditions conducive for biogenic $CH_4$ production" can be changed to "conditions where biogenic $CH_4$ production and potential atmospheric release is likely to occur."

L56f: This assumption is an oversimplification (see above)
Reply: "Intermediate values may indicate gases which are a mixture of biogenic and thermogenic $CH_4$. Complexities can result from geochemical processes such as $CH_4$ oxidation which can change the $\delta^{13}C$-$CH_4$ due to a preference for bacteria to oxidize $CH_4$ containing the lighter isotope ($^{12}C$) enriching the remaining $CH_4$ with $^{13}C$."

Can be changed to:

"Intermediate values may result from the oxidation of biogenic $CH_4$ or from gas which contains a mixture of biogenic and thermogenic $CH_4$. Oxidation of $CH_4$ can occur if gas migrates through an oxidizing environment such as the aerobic zone of the soil or a wetland. This can result in a higher $\delta^{13}C$-$CH_4$ signature due to a preference for bacteria to oxidize $CH_4$ containing the lighter isotope ($^{12}C$), enriching the remaining $CH_4$ with $^{13}C$ (Chanton et al., 2005)."

L85: Do mixed sources contain other $CH_4$ than biogenic and thermogenic? Please clarify.
Reply: Mixed sources contain a mixture of thermogenic and biogenic methane. In order to more clearly identify the intent of the manuscript the final sentence of the introduction can be changed to read "We hypothesise that the largest hotspots in the MRD include contributions of biogenic $CH_4$ due to the abundance of environmental settings where modern methane is being produced (Cunada et al., 2021)."

L159f: The sampling of surface gas with the aluminium tubing is unclear to me. How was the tube filled and how it was possible to analyse discrete samples from this tube? Please explain in more detail.
Reply: The samples collected in aluminium tubing were intended to give a continuous transect of $CH_4$ and $CO_2$ concentrations over the hotspot and were not analyzed as discrete samples. The description of the methods can be rewritten as follows in order to clarify:

"Walking transects were carried out by filling a 30 m coil of 4 mm inside diameter aluminium Synflex tubing while walking at a steady pace across the ground. A constant flow rate of 20 standard cubic centimetres per minute (CCM) was maintained by attaching a small pump and a flow controller to the coil of tubing. Samples were analyzed using a Picarro G2210i analyser immediately on return from the field site. Five walking transects using Synflex tubing took approximately 20 minutes each to fill and covered a distance between 600-800 m. Methane and $CO_2$ concentrations were measured every 1-2 seconds on the air samples collected in aluminium tubing. This allowed for consideration of spatial variability in methane concentrations at each site. Mixing of the air sample inside the tube between collection and analysis is limited due to the small diameter of the tubing. A similar method was used during drone-based $CH_4$ measurements (Andersen et al., 2018)."

L 177: value not ratio
Reply: Ratio can be changed to value

L 275f: This might just indicate a higher contribution of $CH_4$ oxidation in summer than in winter, when the surface soil is frozen.
Reply: We have stated that the difference in $^{13}C$ source signatures could be caused by greater oxidation at this site. The existing text can be changed to the following to make that clearer:

"Discrete sampling at Wetland 1 yielded a $\delta^{13}C$-$CH_4$ Keeling plot source signature of -88.3 ‰ when sampled in October during freeze up and -53.4 ‰ during the summer. We conclude that

this demonstrates a biogenic source during the fall since biogenic production can persist late into the cold season (Zona et al., 2016). While the sampling was carried out at the same location, methane ebullition was seen while sampling during the fall, but not during the summer. The Wetland 1 site was dominated by sedge vegetation with areas of standing water. The lack of ebullition flux at the same site during the summer and the different Keeling plot estimate suggests methane flux in this wetland setting varies seasonally. The Keeling plot source signature of -53.4 ‰ during the summer could be caused from either oxidation of a biogenic source or contributions of both biogenic and thermogenic sources. Oxidation of $CH_4$ has been shown to be a significant source of fractionation in arctic lakes during the summertime (Thompson et al., 2016)."

L282f: Methane oxidation in permafrost-affected wetlands is most important in the ice- free summer. High $CH_4$ oxidation might even cause the lack of ebullition and explain the high $\delta^{13}C$ value of $CH_4$.
Reply: This paragraph can be rewritten to explain the potential for oxidation during the summer (see response to previous comment).

L306: It is unclear, which data indicate the multiple sources of $CH_4$. Please clarify.
Reply: "Flux chamber sampling over the terrestrial shrub tundra terrain immediately adjacent to Lake 1 indicated it was a source of $CH_4$ and $CO_2$ with flux rates of  2.25 mg $CH_4$-C $hr^{-1}m^{-2}$ and 52.73 mg $CO_2$-C $hr^{-1}m^{-2}$. The soil adjacent to Lake 1 was saturated with water, creating ideal conditions for biogenic production at the site. This shows that we sampled multiple sources of $CH_4$ at the site; biogenic methane production in and around the lake as well as a strong thermogenic seep. This could account for the low $r^2$ value (0.48) (Fig. 3) at Lake 1 despite highly elevated $CH_4$."

Can be changed to:

"The soil adjacent to Lake 1 was saturated with water, creating ideal conditions for biogenic production at the site. It is possible that we sampled multiple sources of $CH_4$ at the site; biogenic methane production in and around the lake as well as a strong thermogenic seep. This could account for the low $r^2$ value (0.48) (Fig. 3) at Lake 1 despite highly elevated $CH_4$.

Thermogenic seeps to the atmosphere have been documented in Arctic regions, including the Mackenzie River Delta, where thermogenic methane exists underneath the permafrost (Bowen et al., 2008; Osadetz & Chen, 2010; Walter Anthony et al., 2012). While biogenic $CH_4$ production typically results in $^{13}CH_4$ values less than -70 ‰, oxidation can result in a higher $\delta^{13}C$-$CH_4$ signatures which are closer to that of thermogenic $CH_4$. This is due to a preference for bacteria to oxidize $CH_4$ containing the lighter isotope ($^{12}C$), enriching the remaining $CH_4$ with $^{13}C$ (Chanton et al., 2005). Values of $\delta^{13}C$-$CH_4$ signatures as high as -44.7 ‰, which were observed at Lake 1, are not likely to be generated through oxidation of biogenic $CH_4$. Values almost as high are exceedingly rare at these latitudes, but have been observed before, in an Arctic lake (-49.2 ‰) (Thompson et al., 2016), and in a pond formed in polygonal tundra (-44.9 ‰, -52.3 ‰) (Preuss et al., 2013; Vaughn et al., 2016). In the case of Preuss et al.

(2013), almost complete oxidation of $CH_4$ to atmospheric levels was required to increase the $\delta^{13}C$-$CH_4$ signature from less than -50 ‰ up to -44.9 ‰. Additionally, these two sites are at a river channel and a lake, respectively, where oxidation would be minimal as compared to a wetland. The likelihood of the permafrost thawing completely through at these two locations is also higher than at the wetland locations, increasing the possibility of thermogenic migration."

L311 f: The second part of this sentence is unclear.
Reply: We stand by our assessment that at least 2 of these sites had $\delta^{13}C$-$CH_4$ signatures that indicate thermogenic origin. The values of -42.5 ‰ and -44.7 ‰ are higher than the proven range for biogenic methane in Arctic environments, even sites where there is fractionation due to excessive $CH_4$ oxidation.

We can still reworded this sentence to be clearer, as follows:

"Estimates of source isotope signatures from field sites varied substantially from biogenic to thermogenic, indicating that the largest sites of $CH_4$ production in the MRD are caused by a variety of sources." Can be changed to read: "Estimates of source isotope signatures from field sites ranged from -42 ‰ to -88 ‰, indicating that the largest sites of $CH_4$ production in the MRD are caused by both biogenic and thermogenic sources."

L318f: What are 'eddy covariance hotspot locations' and which data of this study verify these?
Reply: The airborne eddy covariance hotspot locations are described and cited (Kohnert et al., 2017) in the first paragraph of the Methods section and are further detailed in Figure 2.

**Cited literature:**

Bowen, R. G., Dallimore, S. R., Côté, M. M., Wright, J. F., & Lorenson, T. D. (2008). *Geomorphology and gas release from pockmark features in the Mackenzie Delta, Northwest Territories, Canada.* 6.

Bréas, O., Guillou, C., Reniero, F., & Wada, E. (2001). The Global Methane Cycle: Isotopes and Mixing Ratios, Sources and Sinks. *Isotopes in Environmental and Health Studies*, *37*(4), 257–379. https://doi.org/10.1080/10256010108033302

Chanton, J., Chaser, L., Glasser, P., & Siegel, D. (2005). Carbon and Hydrogen Isotopic Effects in Microbial, Methane from Terrestrial Environments. In *Stable Isotopes and Biosphere Atmosphere Interactions* (pp. 85–105). Elsevier. https://doi.org/10.1016/B978-012088447-6/50006-4

Chanton, J., & Smith, L. (1993). Seasonal variations in the isotopic composition of methane associated with aquatic macrophytes. In R. S. Oremland (Ed.), *Biogeochemistry of Global Change: Radiatively Active Trace Gases* (pp. 618–632). Springer US. https://doi.org/10.1007/978-1-4615-2812-8

Conrad, R. (2005). Quantification of methanogenic pathways using stable carbon isotopic signatures: A review and a proposal. *Organic Geochemistry*, *36*(5), 739–752. https://doi.org/10.1016/j.orggeochem.2004.09.006

Fisher, R. E., France, J. L., Lowry, D., Lanoisellé, M., Brownlow, R., Pyle, J. A., Cain, M., Warwick, N., Skiba, U. M., Drewer, J., Dinsmore, K. J., Leeson, S. R., Bauguitte, S. J.-B., Wellpott, A., O'Shea, S. J., Allen, G., Gallagher, M. W., Pitt, J., Percival, C. J., … Nisbet, E. G. (2017). Measurement of the $^{13}$C isotopic signature of methane emissions from northern European wetlands: Northern Wetland CH$_4$ Isotopic Signature. *Global Biogeochemical Cycles*, *31*(3), 605–623. https://doi.org/10.1002/2016GB005504

Happell, J. D., Chanton, J., & Showers, W. S. (1994). The influence of methane oxidation on the stable isotopic composition of methane emitted from Florida swamp forests. *Geochimica et Cosmochimica Acta*, *58*, 4377–4388.

Kohnert, K., Serafimovich, A., Metzger, S., Hartmann, J., & Sachs, T. (2017). Strong geologic methane emissions from discontinuous terrestrial permafrost in the Mackenzie Delta, Canada. *Scientific Reports*, *7*(1), 5828. https://doi.org/10.1038/s41598-017-05783-2

Nakagawa, F., Yoshida, N., Nojiri, Y., & Makarov, VladimirN. (2002). Production of methane from alasses in eastern Siberia: Implications from its $^{14}$C and stable isotopic compositions: METHANE FROM ALASSES IN EASTERN SIBERIA. *Global Biogeochemical Cycles*, *16*(3), 14-1-14–15. https://doi.org/10.1029/2000GB001384

Oh, Y., Zhuang, Q., Welp, L. R., Liu, L., Lan, X., Basu, S., Dlugokencky, E. J., Bruhwiler, L., Miller, J. B., Michel, S. E., Schwietzke, S., Tans, P., Ciais, P., & Chanton, J. P. (2022). Improved global wetland carbon isotopic signatures support post-2006 microbial methane emission increase. *Communications Earth & Environment*, *3*(1), 159. https://doi.org/10.1038/s43247-022-00488-5

Osadetz, K. G., & Chen, Z. (2010). A re-evaluation of Beaufort Sea-Mackenzie Delta basin gas hydrate resource potential: Petroleum system approaches to non-conventional gas resource appraisal and geologically-sourced methane flux. *Bulletin of Canadian Petroleum Geology*, *58*(1), 56–71. https://doi.org/10.2113/gscpgbull.58.1.56

Preuss, I., Knoblauch, C., Gebert, J., & Pfeiffer, E.-M. (2013). Improved quantification of microbial CH$_4$ oxidation efficiency in arctic wetland soils using carbon isotope fractionation. *Biogeosciences*, *10*(4), Article 4. https://doi.org/10.5194/bg-10-2539-2013

Thompson, H. A., White, J. R., Pratt, L. M., & Sauer, P. E. (2016). Spatial variation in flux, $\delta^{13}$C and $\delta^2$H of methane in a small Arctic lake with fringing wetland in western Greenland. *Biogeochemistry*, *131*(1–2), 17–33. https://doi.org/10.1007/s10533-016-0261-1

Vaughn, L. J. S., Conrad, M. E., Bill, M., & Torn, M. S. (2016). Isotopic insights into methane production, oxidation, and emissions in Arctic polygon tundra. *Global Change Biology*, *22*(10), 3487–3502. https://doi.org/10.1111/gcb.13281

Walter Anthony, K. M., Anthony, P., Grosse, G., & Chanton, J. (2012). Geologic methane seeps along boundaries of Arctic permafrost thaw and melting glaciers. *Nature Geoscience*, *5*(6), Article 6. https://doi.org/10.1038/ngeo1480

Zona, D., Gioli, B., Commane, R., Lindaas, J., Wofsy, S. C., Miller, C. E., Dinardo, S. J., Dengel, S., Sweeney, C., Karion, A., Chang, R. Y.-W., Henderson, J. M., Murphy, P. C., Goodrich, J. P., Moreaux, V., Liljedahl, A., Watts, J. D., Kimball, J. S., Lipson, D. A., & Oechel, W. C. (2016). Cold season emissions dominate the Arctic tundra methane budget. *Proceedings of the National Academy of Sciences*, *113*(1), Article 1. https://doi.org/10.1073/pnas.1516017113

---

## Author Comment (AC3)

Dear Referee # 2,

Thank you for your time and contribution in reviewing this manuscript. Responses to general and specific comments are below.

**Review of Characterization of atmospheric methane release in the outer Mackenzie River Delta from biogenic and thermogenic sources**
**Synopsis:**

The authors present results of ground-based measurements of atmospheric methane concentrations and [13]C isotope signatures made in the Mackenzie River Delta, NWT, CN in 2019 and 2021 near locations which had anomalously high fluxes during airborne eddy covariance surveys in 2012 and 2013. They interpret their findings to mean that these previously determined "hotspot" areas were generated either by thermogenic $CH_4$ sources, biogenic $CH_4$ sources, or mixtures of both sources. They conclude that their study only provided a snapshot and that more robust methods are needed to more confidently confirm the origins of the $CH_4$ hotspots identified by the airborne surveys. The authors' dataset is indeed rare, is collected in a climate-sensitive environment, and could be valuable to the scientific community. However, their incomplete methods and lack of description regarding some of the assumptions they make about their sampling approach greatly limit the interpretation of their results. I believe this study will make an important contribution, but not before the authors fully articulate some critical shortcomings about their sampling and whether or not it can actually be used to interpret previously-discovered methane emission hotspots in the Mackenzie River Delta.

**General Comments:**
This study highlights the challenges of relating remotely sensed patterns to those which can be observed from the ground. A proper ground-truthing of remotely sensed signals requires careful consideration of the limitations of both approaches. Aside from a comment that it is impractical to sample the airborne-derived hotspots from the ground, there was little discussion regarding how representative their ground sampling was and whether it could be used to interpret the airborne-derived hotspots at all. The significant spatial disparity between the previous airborne survey and the present ground survey, the minimum spatial resolution at which the airborne survey can be confidently interpreted, and the limited description/presentation of the ground-based survey's methods all raise considerable uncertainty in the study's findings. At a minimum, this study should more explicitly (and quantitatively) describe their findings in space relative to the remotely- observed hotspots. For example, can the walking transect data be represented in figure 1?
Were samples collected up-wind or down-wind of the hotspots? ...etc.
How representative are their discrete samples of the airborne-observed hotspots to which they attempt to attribute their findings? From Figure 1, one can see that the airborne hotspots are several kilometers across. Are the authors attempting to use singular samples from discrete locations to describe $CH_4$ sources from kilometers-wide "hotspots?" The ground-based sampling methods should further discuss the assumptions, strategy, and potential biases involved in their plan. For example, was sampling meant to describe individual features (e.g.

pingos) on the ground or the large-area hotspots more generally? How do the authors reconcile these scale differences? If they intended to apply isotopic signatures and flux rates to individual features on the ground, why not use chambers or other methods to explicitly isolate the air from those sources? If they intended on sampling the air to infer something about the surrounding landscape, why not conduct a more formal wind analysis or "footprint" determination and then describe the features within the footprint? It seems that the approaches fall somewhere in between these two objectives, but as a result, fail to robustly describe the environment in which they were made.

The authors have not clearly articulated their assumptions regarding separating background air from the air that contains mixtures of $CH_4$ originating from their hotspots. Can their methods sufficiently distinguish between the two? How do they know that the background air wasn't already a mixture of other local sources? How do variable wind conditions (velocity **and direction**) affect their sampling approach? There is little description regarding the direction of the wind and how this could affect their goal to interpret the hotspots.

No reference is made to Figure S1 or Figure S2 in the manuscript text. This was surprising, especially since Figure S2 contains the walking transect data.

There are several biases outlined here which are not effectively addressed in the manuscript. Formal wind analysis and flux chamber measurements would improve the quality of this data but we were unable to collect these measurements. We attempted to select methods which would help to characterize these large hotspots with limited space, weight and time on the ground due to the remoteness of these sites. We believe the following changes, along with the changes made to specific comments will highlight and address these biases.

Samples were collected upwind and downwind of any features so that we would be certain to pick up background and peak concentrations if the feature were a source. Wind speed and direction is shown of Figure S2, which can be moved to the main manuscript. We have refrained from trying to make correlations with wind speed because we do not have high-rate wind measurements. The following could be added to the end of the first paragraph in the sample collection section of the methods (3.2, line 165):

"Walking transect locations were selected by completing one transect up wind and one down wind of the estimated location of the highest flux concentration, in order to obtain background concentrations at each site. Discrete point samples were taken parallel to each walking transect, 3-5 m further away from the center of the hotspot. This sampling method was designed to cover the most area, with the fewest samples, in the shortest amount of time possible. By using this method we were still only able to cover a small fraction of the hotspot (5-10% by area), so samples were collected around potential sources of $CH_4$, such as pingos or wetlands. This increased the likelihood of pinpointing the source."

The description of the walking transects can be rewritten as follows:

"Walking transects were carried out by filling a 30 m coil of 4 mm inside diameter aluminium Synflex tubing while walking at a steady pace across the ground. A constant flow rate of 20 standard cubic centimetres per minute (CCM) was maintained by attaching a small pump and a flow controller to the coil of tubing. Samples were analyzed using a Picarro G2210i analyser immediately on return from the field site. Five walking transects using Synflex tubing took approximately 20 minutes each to fill and covered a distance between 600-800 m. Methane and $CO_2$ concentrations were measured every 1-2 seconds on the air samples collected in aluminium tubing. This allowed for consideration of spatial variability in methane concentrations at each site. Mixing of the air sample inside the tube between collection and analysis is limited due to the small diameter of the tubing. A similar method was used during drone-based $CH_4$ measurements (Andersen et al., 2018)."

It would be quite difficult to add more information to Figure 1 and still effectively convey it to the reader. Figure S2 can be moved to the main manuscript and the raw data from the walking transects can be added to the supplement.

**Specific Comments:**

Line 39: A more recent/updated reference is available for the global $CH_4$ budget. Saunois, M., Stavert, A.R., Poulter, B., Bousquet, P., Canadell, J.G., Jackson, R.B., Raymond, P.A., Dlugokencky, E.J., Houweling, S., Patra, P.K. and Ciais, P., 2020. The global methane budget 2000–2017. Earth system science data, 12(3), pp.1561-1623.
Reply: "Current global estimates of wetland $CH_4$ flux to the atmosphere range between 153-227 Tg $CH_4$/yr (Saunois et al., 2016). Non-wetland, freshwater systems are also significant contributors of $CH_4$ to the atmosphere on a global scale (Kirschke et al., 2013) which is estimated between 60-180 Tg $CH_4$/yr (Saunois et al., 2016)."

Can be changed to:

"Current global estimates of wetland $CH_4$ flux to the atmosphere range between 101-179 Tg $CH_4$/yr (Saunois et al., 2020). Non-wetland, freshwater systems are also significant contributors of $CH_4$ to the atmosphere on a global scale (Kirschke et al., 2013) which is estimated between 117-212 Tg $CH_4$/yr (Saunois et al., 2020)."

Line 76: Without context, a "hotspot" can have an ambiguous definition. Since 5 mg $CH_4$ $m^{-2}d^{-1}$ is not a particularly high flux in the context of typical wetland/lake emissions, I think that the authors need to emphasize on what spatial scale this would be considered a high flux or a hotspot. This would also help set the stage for placing the new observations in the proper spatial context.
Reply: This sentence can be rewritten to improve context. Please note that the flux rate threshold used to define geologic emission in Kohnert et al (2017) and for inclusion into this study was 5 mg $CH_4$ $m^{-2}h^{-1}$, not 5 mg $CH_4$ $m^{-2}d^{-1}$.

"These authors assumed that these hotspots were primarily of geologic origin ($CH_4$ produced beneath the permafrost, including thermogenic $CH_4$ from natural gas reserves that has the potential to migrate through discontinuities in the permafrost) since the inferred flux rates of the hotspots identified were significantly greater than the maximum values of around 4 mg $CH_4$ $m^{-2}h^{-1}$ detected for biogenic fluxes north of 61°N (Friborg et al., 2000; Sachs et al., 2008; Sturtevant et al., 2012). These fluxes also occurred in the summer period when most lakes were fully oxygenated, reducing biogenic emissions."

Lines 81 - 86: Check verb tenses in this paragraph. It currently reads like this work has not happened yet- as one would write in a proposal.
Reply: The paragraph will be changed to past tense.

Line 102: Bowen et al. (2008) is plural, so probably change "has" to "have" Lines 119 - 121: Results should be moved to the results section.
Reply: "Has" can be changed to "have" and lines 119-121 can be moved to the results section.

Line 125: What does "focused" mean here? It's also misspelled. This sentence could use a rewrite to improve clarity.
Reply: This sentence can be rewritten as follows:

"Four additional sites (Pingo 3, Wetland 1, Lake 1, Channel Seep) were at locations where point source, aquatic seeps where concentrated ebullitions of $CH_4$ flux were seen in open water in the summer, or in holes in newly formed ice in the fall. Channel Seep and Lake 1 were previously known to researchers in the field party, while Pingo 3 and Wetland 1 sites were identified by holes in the ice which were observed from the helicopter while passing overhead during the fall. Discrete point samples were taken for stable carbon isotope ratio ($\delta^{13}$C-$CH_4$) analysis."

Line 155: Could sampling only in wetlands bias the observations towards one emission pathway (or source) over another? Do thermogenic emissions only occur in wetlands and/or lakes? Or can they occur in dry areas as well? Some discussion and communication of assumptions around this sampling strategy are warranted (see general comment).
Reply: While this was the best method to cover very large sites that were accessible only by helicopter. We agree that this sampling method could bias the results toward the obvious features which we sampled. The following can be added to clarify:

"Sampling transect locations were selected within the general hotspot area where the airborne eddy covariance fluxes were highest. Samples were concentrated around features such as wetland areas or pingos that were considered possible sources of $CH_4$. Pingos and wetlands can be sources of both biogenic and thermogenic $CH_4$ but in dry areas biogenic emissions are far less likely than thermogenic emissions. Unfortunately, this has the potential to bias sampling to areas where biogenic emissions are more likely."

Lines 159 – 164: Is this an "air core?" or something else? This description is somewhat vague. It raises several questions. What is the objective of the "walking transect?" How was the tubing filled? What are "analytical determinations?" Mixing of what?

Reply: Yes, these are essentially air cores, these samples were intended to give a continuous transect of $CH_4$ and $CO_2$ concentrations. The description of the methods can be rewritten as follows in order to clarify:

"Walking transects were carried out by filling a 30 m coil of 4 mm inside diameter aluminium Synflex tubing while walking at a steady pace across the ground. A constant flow rate of 20 standard cubic centimetres per minute (CCM) was maintained by attaching a small pump and a flow controller to the coil of tubing. Samples were analyzed using a Picarro G2210i analyser immediately on return from the field site. Five walking transects using Synflex tubing took approximately 20 minutes each to fill and covered a distance between 600-800 m. Concentrations of $CH_4$ and $CO_2$ were made every 1-2 seconds on the air samples collected in aluminium tubing which allowed for consideration of spatial variability in methane concentrations at each site. Mixing of the air sample inside the tube between collection and analysis is limited due to the small diameter of the tubing. A similar method was used during drone-based $CH_4$ measurements (Andersen et al., 2018)."

Line 168: The aerial EC hotspot sites? Or on walking transects, or both? Be more specific about the spatial representativeness of the observations.

Reply: The following could be added to the end of the first paragraph in the sample collection section of the methods (3.2, line 165): "Walking transect location was selected by completing one transect up wind and one down wind of the estimated location of the highest flux concentration, in order to obtain background concentrations at each site. Discrete point samples were taken parallel to each walking transect, 3-5 m further away from the center of the hotspot. This sampling method was designed to cover the most area, with the fewest samples, in the shortest amount of time possible. By using this method we were still only able to cover a small fraction of the hotspot (5-10% by area), so samples were collected around potential sources of $CH_4$, such as pingos or wetlands. This increased the likelihood of pinpointing the source."

Line 175: How were fluxes derived from chamber data? How many vials were analyzed per flux observation? For how long? Linear extrapolation? Was non-linearity observed? If so, how was this handled in data QA/QC? These parameters are typically reported alongside chamber flux data.

Reply: We had originally included the flux chamber measurements because they have considerable value. There is a lack of published chamber measurements in the Mackenzie River Delta as well as the Canadian Arctic. The chamber flux data will be removed from the manuscript since it does not add significantly and distracts from the main narrative presented in this manuscript.

Line 221: What is "reasonable confidence?" Can you be more quantitative? Correlation statistic? Otherwise, this is not very informative.

Reply: The sentence on line 221 can be changed to the following:

"Estimates of source stable carbon isotope signatures ($\delta^{13}C$-$CH_4$) derived from Keeling plots were -53.0 (+/- 1.01) ‰ for Pingo 1 and -63.6 (+/- 1.87) ‰ for Pingo 2."

Line 222: Phrases like "substantially enriched" and "relatively close" are not very informative and weaken the impact of the claims being made. Please quantify where possible.

Reply: The sentence on line 222 can be changed to the following:

"These signatures are enriched in $^{13}C$ compared to typical biogenic sources with $\delta^{13}C$-$CH_4$ < -70 ‰ and within 14 ‰ of our assumed threshold of -50 ‰ based on thermogenic gas found in the Taglu hydrocarbon reservoir."

Line 266: How was "background" determined?

Reply: Site 9 data can be added to Table 1. Background was determined to be 1.995 ppm $CH_4$, which was the average value recorded at the ECCC weather station at Inuvik during the time of the fieldwork.

"No elevated atmospheric methane values were detected during walking transects at Site 9" can be changed to:

"Atmospheric $CH_4$ values during walking transects at Site 9 peaked at 2.044 ppm,"

**Cited Literature:**

Andersen, T., Scheeren, B., Peters, W., & Chen, H. (2018). A UAV-based active AirCore system for measurements of greenhouse gases. *Atmospheric Measurement Techniques*, *11*(5), 2683–2699. https://doi.org/10.5194/amt-11-2683-2018

Friborg, T., Christensen, T. R., Hansen, B. U., Nordstroem, C., & Soegaard, H. (2000). Trace gas exchange in a high-Arctic valley: 2. Landscape $CH_4$ fluxes measured and modeled using eddy correlation data. *Global Biogeochemical Cycles*, *14*(3), 715–723. https://doi.org/10.1029/1999GB001136

Kirschke, S., Bousquet, P., Ciais, P., Saunois, M., Canadell, J. G., Dlugokencky, E. J., Bergamaschi, P., Bergmann, D., Blake, D. R., Bruhwiler, L., Cameron-Smith, P., Castaldi, S., Chevallier, F., Feng, L., Fraser, A., Heimann, M., Hodson, E. L., Houweling, S., Josse, B., … Zeng, G. (2013). Three decades of global methane sources and sinks. *Nature Geoscience*, *6*(10), Article 10. https://doi.org/10.1038/ngeo1955

Sachs, T., Wille, C., Boike, J., & Kutzbach, L. (2008). Environmental controls on ecosystem-scale $CH_4$ emission from polygonal tundra in the Lena River Delta, Siberia. *Journal of Geophysical Research*, *113*, G00A03. https://doi.org/10.1029/2007JG000505

Saunois, M., Bousquet, P., Poulter, B., Peregon, A., Ciais, P., Canadell, J. G., Dlugokencky, E. J., Etiope, G., Bastviken, D., Houweling, S., Janssens-Maenhout, G., Tubiello, F. N., Castaldi, S., Jackson, R. B., Alexe, M., Arora, V. K., Beerling, D. J., Bergamaschi, P., Blake, D. R., … Zhu, Q. (2016). The global methane budget 2000–2012. *Earth System Science Data*, *8*(2), Article 2. https://doi.org/10.5194/essd-8-697-2016

Saunois, M., Stavert, A. R., Poulter, B., Bousquet, P., Canadell, J. G., Jackson, R. B., Raymond, P. A., Dlugokencky, E. J., Houweling, S., Patra, P. K., Ciais, P., Arora, V. K., Bastviken, D., Bergamaschi, P., Blake, D. R., Brailsford, G., Bruhwiler, L., Carlson, K. M., Carrol, M., … Zhuang, Q. (2020). The Global Methane Budget 2000–2017. *Earth System Science Data*, *12*(3), 1561–1623. https://doi.org/10.5194/essd-12-1561-2020

Sturtevant, C. S., Oechel, W. C., Zona, D., Kim, Y., & Emerson, C. E. (2012). Soil moisture control over autumn season methane flux, Arctic Coastal Plain of Alaska. *Biogeosciences*, *9*(4), 1423–1440. https://doi.org/10.5194/bg-9-1423-2012

---

## Referee Report (RR1)

Review of 2022-549: "Characterization of atmospheric methane release in the outer Mackenzie River Delta from biogenic and thermogenic sources." by Wesley et al.

Wesley et al. have studied hotspot of methane from the Mackenzie Delta, measuring stable isotopes from atmospheric methane above know aquatic and terrestrial hotspot of methane. Their $\delta^{13}$C-CH$_4$ signatures indicate that both biogenic and thermogenic sources are found in the delta.

I find that study very interesting as methane hotspots are rarely characterized, especially I appreciate the effort of verifying data obtained by airborne eddy covariance. Although the airborne study has been realized in 2013 and the discrete sampling in 2019. In this study they characterized very few hotspots (8) and it would be worthwhile to continue this study and add more data to be able to derive a regional pattern. I recommend for publication with very minor correction.

**General comments:**

Lake 1 is also known under the name "Swiss Cheese lake", it would be useful for the reader not familiar with this area to mention that in the supplement (for example in table S1).

**Main text:**

L168-170: The sentence starting with "Sampling transect locations…"is repeated

L231: There is  a "-" missing before "53"

---

## Referee Report (RR2)

**Review:**
"Characterization of atmospheric methane release in the outer Mackenzie River Delta from biogenic and thermogenic sources" (egosphere-2022-549) Submitted on 26 Jun 2022.

**Reviewer:**
Nicholas R. Hasson (NRH), University of Alaska Fairbanks (no-competing conflicts)

**Assessment:**
The study provides significant insight into the findings of Kohnert et al. (2017), which have spurred a multitude of subsequent airborne and ground measurements across the Arctic, a practice that continues to this day. The results suggest a variety of methane production sources in the MRD, with source isotope signatures ranging from -42 ‰ (biogenic) to -88 ‰ (thermogenic), range values indicates that methane in the MRD is likely being produced by both biogenic and thermogenic sources and suggest some mixing, strongly linked to seasonality. This is perfectly in line with what we know about methane production and the interpretation, which seems scientifically valid, does add important reference results to the limited data from airborne/ground coupled surveys on methane hotspot investigations.

I would like the authors to consider recent work (post-Kohnert, 2017) on characterizing hotspots in MRD has provided significant results (e.g. Elder et al., 2020, 2021; Baskaren et al. 2022). Please consider these more-recent updated references on magnitude and occurrence of methane hotspots in MRD, and how this enhances, supports, or disagrees with your findings. Additionally, consider thermogenic and biogenic isotopic signatures from recent similar work (Kleber et al., 2023; Sullivan et al., 2021; Elder et al., 2018), which may support or not the interpretation concluded here. It may be important to briefly mention how so.

However, these results significantly add to import investigation on the discussion on source attribution (e.g. biogenic vs. thermogenic) sources of methane hotspots, particularly important for airborne validation and coupling of ground truth observation, yet data has remained limited. Therefore, this warrant publications with minor technical adjustments, such as the inclusion of additional supporting reference material that supports or challenges these results and/or provides valuable recent observations (post-Kohnert, 2017) that alludes to the behavior of hotspots in MRD. The comments should be viewed as recommended suggestions for these reference materials, and the authors have the discretion to merely incorporate the reference material without my proposed text modifications. However, I have provided some examples of how the text could be altered to either support or challenge interpretations, which may further enrich these crucial ground truths for ongoing campaigns carrying out similar observations.

In summary, these findings are important and significant results for future endeavors focusing on airborne/ground verification of methane hotspots in the Arctic.

**Review Summary:**

The authors present novel data and likely is the first to measure stable carbon isotope signatures of atmospheric methane at hotspots in the Mackenzie River Delta (MRD). The results suggest a variety of methane production sources in the MRD, with source isotope signatures ranging from -42 ‰ (biogenic) to -88 ‰ (thermogenic). Of the eight sites investigated, two had a thermogenic origin, four were biogenic, and two were possibly a result of oxidation of mixed biogenic/thermogenic sources. This is supported by other recent studies.

These results also suggest methane migration from below the thin permafrost at most sites, including from the Taglu gas field, over an area of approximately 20 km north to south, which suggest complex permafrost distribution (e.g. due to the hydrology of river taliks, pingo-systems, coastal settings, and together a mix of likely through-taliks and permafrost degradation). The study was able to validate airborne eddy covariance hotspot locations using walking transects to measure atmospheric methane variation. However, authors point out these methods only provide a snapshot of methane sources during site visits, and a comprehensive understanding of annual methane production is yet to be established.

Authors suggest that future research should include year-round flux measurements and stable carbon isotope measurements to fully quantify the annual methane emission from both biogenic and thermogenic sources. Additionally, combining portable methane analyzers with flux chamber and isotopic measurements could help to better identify and quantify sources, particularly at sites where both biogenic and thermogenic sources are likely. Geophysical mapping atop these transects would provide a useful coupling to permafrost distribution and potential source attribution between biogenic and thermogenic sources.

I highly recommend adding some of the added reference support material **(#1-13)** and considering comments #1-23. Particularly important, **reference material post-Kohnert et al. results from MRD, e.g., Elder et al. (2020, 2021) and Baskaren et al. (2022).** I would like the **authors to consider (Kleber et al., 2023; Sullivan et al., 2021; Elder et al., 2018)**, which may serve to challenge or support these results, and it may be important briefly mentioning how so.

I hope these added suggestions and reference materials can serve to enhance your otherwise significant results. Nice work.

**Review by section:**

**1    Introduction    -    Please    consider    comments/suggestions    #1-11**

**#1**NRH: please consider additional reference supports (A,B), which updates the magnitudes of methane hotspots (including from MRD) and the spatial/temporal distribution, which have been found to follow a power law series as a function of distance to stand water; arctic hotspot methane law (e.g. <40 m from wetland boundary).

**#2**NRH: Furthermore, extreme hotspots have been shown to range from **48 mg to 1008 mg CH$_4$ m$^{-2}$ hr$^{-1}$**, including from MRD region. However, I mentioned from a recent meta analysis on terrestrial sources of methane (e.g. >60 N)(Kuhn et al, 2021) shows the roughly average yearly CH$_4$ emission from terrestrial permafrost areas to be 2.22 mg m$^{-2}$ hr$^{-1}$. Therefore, your reported {4-5 mg m$^{-2}$ h$^{-1}$ values} **are considered high or roughly double the average hourly mean.**

**#3**NRH**: I would label extreme hotspots,** including from MRD region, to include the sources found more recently (post-Kohnert study) by Elder et al. (2020,2021) and Baskaren et al. (2022); the work here highlights the importance of your work and significant of the Kohnert et al. follow up (e.g. isotopic fingerprinting associated with hotspots), but currently, lacks these updates references. I've organized potential text below:

A.  Additional refrences for spatial and topographic methane hotspots in MRD

**#4**NRH: I highly recommend adding recent reference support regarding methane hotspot detection in Mackenzie Delta Region (MRD). I've added a potential way to include citation in the text (line 91-95)

Suggested text inclusion
"In the MRD, arctic CH$_4$ hotspots have been found to exhibit a power law relationship with the distance to the nearest standing water (Elder et al., 2020). The geomorphic factors controlling CH$_4$ hotspots in the MRD reveal a spatial decay in the correlation between distance to water and land cover or vegetation type (Baskaran et al., 2022).

Suggested supporting reference #1:
Elder, C. D., Thompson, D. R., Thorpe, A. K., Hanke, P., Walter Anthony, K. M., & Miller, C. E. (2020). Airborne mapping reveals emergent power law of Arctic methane emissions. Geophysical Research Letters, 47, e2019GL085707. https://doi.org/10.1029/2019GL085707

Suggested supporting reference #2:
Baskaran, Latha & Elder, Clayton & Bloom, A. & Ma, Shuang & Thompson, David & Miller, Charles. (2022). Geomorphological patterns of remotely sensed methane hot spots in the Mackenzie Delta, Canada. Environmental Research Letters. 17. https://dx.doi.org/10.1088/1748-9326/ac41fb

B. Additional references for methane hotspot magnitudes

**#5**NRH**::** Please add updated reference to the context of "extreme methane hotspots" equating to {4-5 mg m$^{-2}$ h$^{-1}$}. For example, Elder et al. (2021) reported "Ground-based chamber measurements confirmed average daily CH$_4$ fluxes of 1,170 mg m$^{-2}$ d$^{-1}$, with extreme daily maxima up to 24,200 mg CH$_4$ m$^{-2}$ d$^{-1}$". Converting to hourly, this equates to **48 mg to 1008 mg m$^{-2}$ hr$^{-1}$**. **Which are considered extreme.**

Suggested supporting reference #3:
Elder, C. D., Thompson, D. R., Thorpe, A. K., Chandanpurkar, H. A., Hanke, P. J., Hasson, N., et al. (2021). Characterizing methane emission hotspots from thawing permafrost. *Global Biogeochemical Cycles*, 35, e2020GB006922. https://doi.org/10.1029/2020GB006922

**#6**NRH**:** line 94-96 from your text, I recommend adding more recent hotspot values, e.g.,...

… "hotspots identified {were high, when considering the} maximum published value of around 4 mg CH$_4$ m$^{-2}$ hr$^{-1}$ detected for biogenic fluxes north of 61°N (Friborg et al., 2000; Sturtevant et al., 2012; Sachs et al., 2008).

**#7**NRH**:** <additionally> …{However, recent hotspot observations from >60°N show more extreme hotspots ranging from **48 mg to 1008 mg CH$_4$ m$^{-2}$ hr$^{-1}$** , including from the MRD (Elder et al., 2020; 2021; Baskaran et al. 2022)}...<additionally>…{and recent meta analysis shows the **yearly daily average of terrestrial hotspots to range 5.04 to 29.3 mg CH$_4$ m$^{-2}$ h$^{-1}$**, when considering other emission inventories >4 mg **CH$_4$ m$^{-2}$ h$^{-1}$**. }. Futhermore, from a more holistic reference gathering, I made a rough calculation of the terrestrial database of methane emissions (Kuhn et al., 2021), including 105 previous methane flux investigations, I calculated the average CH$_4$ emission to be 53.3 mg m$^{-2}$ d$^{-1}$ (range: 705 to -9.8; mg m$^{-2}$ d$^{-1}$ ;N=545). When converting to hourly, the average yearly CH$_4$ emission from terrestrial permafrost areas showed 2.22 mg m$^{-2}$ hr$^{-1}$.

**#8**NRH**:** Therefore, your reported {4-5 mg m$^{-2}$ h$^{-1}$ values} **are considered high or roughly double the average hourly mean.** However, under the context of eddy covariance open-air mixing values, which are very high in this context (e.g. versus chamber-derived emissions).

**#9**NRH**:** I would mentioned this. >Values can be higher, with recently reported the yearly daily average of 705 to 121 mg m$^{-2}$ d$^{-1}$ or ranging 5.04 to 29.3 mg m$^{-2}$ h$^{-1}$. Together with the >60°N extreme hotspots ranging from **48 mg to 1008 mg CH$_4$ m$^{-2}$ hr$^{-1}$** , including from the MRD (Elder et al., 2020; 2021; Baskaran et al. 2022)}

Suggested supporting reference #4:
Elder, C. D., Thompson, D. R., Thorpe, A. K., Chandanpurkar, H. A., Hanke, P. J., Hasson, N., et al. (2021). Characterizing methane emission hotspots from thawing permafrost. *Global Biogeochemical Cycles*, 35, e2020GB006922. https://doi.org/10.1029/2020GB006922

Suggested supporting reference #5:
McKenzie Kuhn, Ruth Varner, David Bastviken, Patrick Crill, Sally MacIntyre, et al. 2021. BAWLD-CH4: Methane Fluxes from Boreal and Arctic Ecosystems. Arctic Data Center doi:10.18739/A2DN3ZX1R.

**#10**NRH**:** Note, the isotopic composition of $CH_4$ hotspots characterized in Elder et al. (2021), was later investigated and showed biogenic origin: "**$CH_4$ hotspot revealed a $^{14}C$ age of 35,360 YBP ($\delta^{13}C$ -73.8 ‰)**(Hasson et al. 2022)." This supports your biogenic source attributions of hotspots detected by airborne observations, although from Alaska.

Suggested supporting reference#6
Hasson, N., et al. (2022). Methane emissions show exponential inverse relationship with electrical resistivity from discontinuous permafrost wetlands in Alaska. AGU Fall Meeting Abstracts. AAGUFM.B15E..06H https://ui.adsabs.harvard.edu/abs/2022AGUFM.B15E..06H

**#11**NRH**:** Note, it may also be useful to discuss the Elder et al. (2021) results, which goes on to upscale these observations from arctic and pan-arctic hotspot detection, includeing MRD:

e.g., "Emissions from the hotspot accounted for ~40% of total diffusive CH4 emissions from the entire study area. Combining these results with hotspot statistics from our 70,000 km$^{-2}$ airborne survey across Alaska and northwestern Canada (e.g. MRD), we estimate that terrestrial thermokarst hotspots currently emit 1.1 (0.1 – 5.2) Tg CH4 yr-1, or roughly 4% of the annual pan-Arctic wetland budget from just 0.01% of the northern permafrost land area." This support MRD hotspot signficance.

No further comments.

**Section 2:** Setting - Please consider adding reference material for context/support **#12-13**

**#12**NRH: I recommend the additional references to support your setting, e.g., at line #116-118:

< "Permafrost is generally seen as being continuous under land areas in MRD, but it is mostly missing under lakes that don't freeze all the way to the bottom during winter (Nguyen et al., 2009)." <adding further support context> "Additionally, it has been demonstrated in permafrost regions >60 N, **river taliks extend beyond the river plane by connecting through-taliks with nearby wetlands** (Minsley et al., 2012; see Figs 4,5), showing how "discontinuous" permafrost conditions could be present adjacent to MRD tributaries">

**#13**NRH: **why not include** something like <"The airborne geophysical data detailed in Minsley et al. (2012), illustrates the complexity of river through-taliks and hydrology (e.g. Yukon river) which may mimic areas like the Mackenzie River Delta (MDR). This complexity leads to the presence of discontinuous permafrost near rivers, indicating that the MDR might be fragmented near river tributaries, allowing gas-conduits to form along hydrological through-taliks. These intricate river

taliks have the potential to connect with lakes, as observed in the Yukon and Noatak rivers in Alaska, and have been identified in close proximity to some of the largest geological seep sources (Sullivan, 2021). Recent studies have further uncovered that methane-rich groundwater springs are transporting deep thermogenic and biogenic methane gas, such as in Svalbard, north of 79°N latitude. This gas reaches the surface and is found to be supersaturated with methane at levels up to 600,000 times greater than what would be expected for equilibrium with the atmosphere (Klebar et al., 2023).">...

- For example, river talik networks can fragment "continuous permafrost" into "discontinuous permafrost" by through-taliks near large watersheds (e.g. MDR), which may help support why hotspots of deeper thermogenic emissions would occur here in "continuous zone permafrost".

- For example, Sullivan et al. shows "microbially produced fossil CH4 is being vented though a narrow thaw conduit below Esieh Lake through pockmarks on the lake bottom. This is one of the highest flux geologic CH4 seep fields known in the terrestrial environment and potentially the highest flux single methane seep", and is along the Noatak river, suggesting the river talik is at work here.

- For example: recent work (2023) highlights the context of your investigation on thermogenic sources in settings section, e.g., : "methane-rich groundwater springs are bringing deep-seated methane gas to the surface. Waters collected from these springs during are supersaturated with methane up to 600,000 times greater than atmospheric equilibration. Spatial sampling reveals a geological dependency on the extent of methane supersaturation, with isotopic evidence of a thermogenic source. Waters collected from these springs are supersaturated with methane up to 600,000 times greater than atmospheric equilibration.

Suggested supporting reference #7-9

Minsley, B. J., et al. (2012), Airborne electromagnetic imaging of discontinuous permafrost, Geophys. Res. Lett., 39, L02503, doi:10.1029/2011GL050079.

Sullivan, TD, Parsekian, AD, Sharp, J, et al. Influence of permafrost thaw on an extreme geologic methane seep. Permafrost and Periglac Process. 2021; 32: 484–502. https://doi.org/10.1002/ppp.2114

Kleber, G.E., Hodson, A.J., Magerl, L. et al. Groundwater springs formed during glacial retreat are a large source of methane in the high Arctic. Nat. Geosci. 16, 597–604 (2023). https://doi.org/10.1038/s41561-023-01210-6

No further comments.

**Section 2:** Study Location - Please consider reference support **#14**

**#14**NRH :Line 175 reference suggestion (context)…e.g., **context for the geophysical extent of through-taliks near sites (1) channel seep, Pingo 2, Pingo 1, and Site 9…**

>"However, nearby reports along the arctic coastal shelf have indicated the lack of ice-bonded permafrost beneath the coastline, implying the existence of extensive through-taliks that intrude sea-to-land. These through-taliks are potentially connecting subpermafrost aquifers on land, alluding to the poorly understood and highlight complex dynamics of coastal subsurface hydrology."> impacting potentially (1) channel seep, Pingo 2, Pingo 1, and Site 9…

Suggested supporting reference #10

Micaela N. Pedrazas et al., Absence of ice-bonded permafrost beneath an Arctic lagoon revealed by electrical geophysics.Sci. Adv.6,eabb5083(2020).DOI:10.1126/sciadv.abb5083

No further comments.

**Section 3:** Methods, Sample collection and analysis - comments/suggestions **#14-15**

Please consider added reference to warrant justification of sampling protocols on hotspots, which support authors approach, which does support initial hypothesis (e.g. distance from standing water=larger source attribution target).

NRH: The authors used similar method is based on protocols by Andersen et al., 2018. Authors point out that this method are only able to recover a small fraction of the hotspot (e.g. 5-10% by area). Note, as previous aforementioned studies allude to (e.g. Elder et al., 2021, Hasson et al., 2022), even though hotspots can dominate the local diffusive CH4 budget (e.g. 90% total), these arise from only a tiny fraction of the area  (e.g. 15% of observations), suggesting that although Kohnert et al. observations show large coverage (e.g. 1 km2), these hotspots may come from discrete areas (e.g. 150-200 m transects), suggesting this study may have captured these disproportionally large sources from a disproportionally small area, similar to Elder et al. 2021

**#14**NRH: The importance of these sampling protocols rest on the hypothesis that the sample locations chosen (e.g. near wetlands and pingos, etc.) are the source of these hotspots. **This is not necessarily unwarranted bias,** since prior work has shown that hotspots follow a power law series from distance to standing water (e.g. 0-100 m)(e.g. Elder et al. 2020, 2021; Baskaran et al. (2022)). Its interesting that the hotspots from pingos are near the lowlands (e.g. supporting Baskaran et al., analysis of topographical controls). Given the considerable expensive of field work by helicopter and challenging environment, which limits sampling observational time, future work may also consider transects <100 m from wetlands or areas with complex through-taliks and complex permafrost hydrology (e.g. Pingos, tributaries, coastline).

The authors may wish to highlight this fact for MRD hotspots to minimize effects of bias or rational for the interpretation of data.

Therefore, if sampling protocols reference the power law findings by Elder et al., 2020,Baskaran et al. 2022, the bias near wetlands has a justified reason. Furthermore, discrete changes from upland to lowlands (escarpment bluffs, river terraces, lake terraces, etc.) have been shown to be statistically related to high occurrence of MRD hotspots (Baskaran et al. (2022)). **The authors sampling protocol near distance to standing water or topographic changes near targets is a good approach.**

**#15**NRH: Suggestion, it might be advantages to show the sampling transects as a function of distance to standing water (e.g. Channel seep was ~50 m from standing water, etc.). Does this spatial relationship tell us anything more about the data (e.g. isotopic signature, concentrations?). Perhaps the authors can mention any relationship that is found, e.g. similar to the pingo lowland having higher concentration than pingo upland, so future studies can plan sampling around these known spatial relationships between geomorphology and water table position.

No additional comments on methods here.

**Section 3:** Methods, Determination of CH4 source stable isotope value - **#16-17**

NRH: Authors justify mass balance approach, considering limitations and assumptions. In this context, sampling minimized bias by sampling upwind and downwind of the estimated locations, along 600-800 m transects, which provided a large spatial and temporal range estimate of source attributions. Wind regime was observed using handled anemometer with accuracy of +/- 0.1 m/s. From the paragraph (L239-256), the authors acknowledge the limitations of the Keeling plot analysis under certain field conditions. They accept that the method's assumption of only two components being measured (the source released at the surface/atmosphere interface and the background regional atmospheric signature) can be challenging in broad areas or windy conditions that may cause mixing.

Given these constraints, its acceptable to assume these limitations when applying the Keeling plot analysis, which depend on the specific context of the hypothesize: the authors clarify earlier that hotspots may form from discrete areas near wetlands and pingos. It seems than the significance of the data is determined by the wind direction from source target (wind direction from or away source target). The authors have acknowledged the limitations and adapted their methodology, accordingly, attempting to collect samples as close to the known point source of emissions as possible when observable ebullition was present.

Their acceptance of the limitation when appraising large hotspots, based on their assumption that the atmospheric point samples taken within these hotspot source regions could represent a mixed δ13C-CH4 signature, seems a reasonable approach given the constraints of field conditions and the necessity to identify broad trends.

**#16**NRH: I recommend complementary rose wind plots that show the wind direction versus source target over time, and how that may contribute to the interpretations of results. Showing wind rose plots (over time, e.g. 30 minutes) shows how the upwind effects total wind mixing effects on dilute

concentrations or may allude to the source (e.g. increased as wind shifted NE, versus SW). Although, currently, the authors do show in Figure 3 the wind direction (red arrow vector), **but perhaps a wind rose representing the time domain of sampling could be shown.** Although, not necessary, if this "red arrow" on wind represents a mean over the duration of time sampled or constant direction.

**#17**NRH: **Does the red vector arrow show the mean or constant direction of wind over the sampling period?** If so, mentioned this in Figure 3 caption (e.g. wind direction represents average during sampling time).

No further comments.

**Section 4**: Results

NRH: In summary, the results show trends of elevated methane and carbon dioxide concentrations in specific sites and a variability in stable carbon isotope signatures, indicating possible differences in the sources of these greenhouse gasses at each site. The results highlight the observation of elevated atmospheric CH4 concentrations at four of the five (Pingo 1, Pingo 2, Wetland 2, Wetland 3) where walking transects were performed.

The results (as expected, from hypothesis) show that proximity to source of seep, e.g., ebullition (closest to standing water) which had substantially elevation methane concentrations. From this, I am curious of the relationship between distance to standing water and/or topographic upland versus lowland relationship to both concetrations and isotopic signatures elsewhere. Also noteworthy, Keeling plot values show -88.4‰ for wetlands in the fall, during maximum thaw, or when thermal lag from summer months penetrates deeper in talik sources, given the latency heat of water. Whereas Keeling plot values show -56.7‰ during summer months, when the cold season thermal lag can be substantial in the subsurface.

**#18**NRH: Perhaps this suggest age dependency on talik versus time of thermal lag in subsurface and seasonal thermal lags associated with age. Alternatively, hydrology or seasonality of groundwater surge (higher in summer, lower in august) may result in more mixing effects. It might be useful to discuss this later.

Line 273-274: > Keeling plot values were -88.4‰ for Wetland 1 when sampled in the fall, but -56.7 ‰ when sampled in the summer.>

**#19**NRH: Please add month (x), e.g., fall (X), summer (X) in Line 273-274.

No further comments.

**Section 5.1:** Results - Consider comments on reference material #20-21

- Outer delta pingos

NRH: The study discovered high methane concentrations in airborne hotspots in the northwestern outer MRD. The carbon isotope signatures from these sites suggest a likely mixture of thermogenic and biogenic methane sources. Despite the proximity of these high readings to pingo features, notably, **the highest values were detected in the surrounding low-lying shrub tundra, not the features themselves.**

**#20**NRH: This suggest topographical relationships to hotspots or water table position by Elder et al., Baskaren et al., etc.

NRH (Line 315): indeed, I agree, this is exciting result and assumptions about the permafrost through-taliks, pingo hydrology, and complex permafrost and absence of permafrost does warrant geophysical transects. It seems understanding the source attribution is a function of the assumptions about permafrost or absence of permafrost, which can be demarcated by low-frequency EM geophysical transects.

- Wetland sites

<sampling at Wetland 1 revealed a seasonal variability in the carbon isotope signature, with a value of -88.3‰ in October (suggesting a biogenic source) and -53.4‰ during the summer (indicating a mixed source). This could be due to methane oxidation or a blend of biogenic and thermogenic sources.> >The lack of observed methane ebullition during summer suggests a potential seasonal variation in methane flux in these wetland settings.>Therefore, contributions to the atmospheric methane at Wetland 1 during summer could be from both biogenic and thermogenic sources, with oxidation and varying production pathways also being plausible.>

>Similar observations for seasonal variability in terrestrial sources are not well documented in the literature, although transport of CH4 from anaerobic soils with sedge vegetation has been observed to bypass the aerobic zone, limiting oxidation during the growing season (Olefeldt et al., 2013; King et al., 1998). Therefore, it is possible that there were contributions to the atmosphere from biogenic and thermogenic sources at Wetland 1, but oxidation and varying production pathways cannot be ruled out as the reason for the signature derived during the summer sampling>

**#21**NRH:Interesting result. This may also allude to thermal lag times moving through different depths seasonally or perhaps more important here,  the mixing blend of biogenic and thermogenic sources are enhanced by summer peak water levels, which can transport various methane sources significantly in the watershed. **Groundwater discharge  as a driver of methane emissions and mixing from Arctic lakes and transporting methane from upland active layer thaw to lowland area has been shown to be significant,** supporting oxidation and blending of biogenic and thermogenic source.

… **Ground water discharge is known to follow as seasonal trend, which perhaps alludes to the shifts from summer to fall data, with greater mixing of biogenic and thermogenic sources in summer, and more stable (deeper) biogenic sources emanating directly from the talik, without ground water mixing.** For example, Please see Olid et al( 2022) and Paytan et al. (2015).

Suggested supporting reference #11
Olid, C., Rodellas, V., Rocher-Ros, G. *et al.* Groundwater discharge as a driver of methane emissions from Arctic lakes. *Nat Commun* 13, 3667 (2022). https://doi.org/10.1038/s41467-022-31219-1

Suggested supporting reference #12
Lecher, Alanna & Dimova, Natasha & Sparrow, Katy & Garcia-Tigreros, Fenix & Murray, Joseph & Tulaczyk, Slawek & Kessler, John. (2015). Methane transport from the active layer to lakes in the Arctic using Toolik Lake, Alaska, as a case study. Proceedings of the National Academy of Sciences of the United States of America. 112. 10.1073/pnas.1417392112.

No further comments.

**Section 5.2:** Results - Consider more recent reference support #21 and counter evidence #22,23 (if applicable), and further recent support for thermogenic versus biogenic signatures in arctic from #24

Line 368: >Some of the largest occurrences of atmospheric release of CH4 in Arctic environments have been reported in association with large gas seeps of thermogenic CH4, causing high-rate ebullition (Walter Anthony et al., 2012)>

**#21**NRH: consider also adding the more recent literature for support for both biogenic sourced geological                        gas                        and                        thermogenic                        >

Kleber, G.E., Hodson, A.J., Magerl, L. et al. Groundwater springs formed during glacial retreat are a large source of methane in the high Arctic. Nat. Geosci. 16, 597–604 (2023). https://doi.org/10.1038/s41561-023-01210-6

Sullivan, TD, Parsekian, AD, Sharp, J, et al. Influence of permafrost thaw on an extreme geologic methane      seep.      Permafrost      and      Periglac      Process.      2021;      32:      484–502. https://doi.org/10.1002/ppp.2114

**#22**NRH: I am curious how these results of δ13C-CH4 source signatures (e.g. -42.0 to 44.7 ‰) challange large gas seeps of thermogenic CH4, when Elder et al. (2018) showed these signatures associated with alaskan lakes dominated by young carbon, which may challenge this interpretation.

For example, you may wish to add this edits:

<Some of the largest occurrences of atmospheric release of CH4 in Arctic environments have been reported in association with large gas seeps of thermogenic CH4, causing high-rate ebullition (Walter Anthony et al., 2012). **However, arctic coastal plain CH4 emissions have been associated with younger carbon sources (Elder et al., 2018)**">

**#23**NRH: you may want to download Elder et al. supp section and take a look at those corresponding δ13C-CH4 signatures and similarities or dissimilarities to your data.

Suggested supporting reference #13

Elder, C.D., Xu, X., Walker, J. *et al.* Greenhouse gas emissions from diverse Arctic Alaskan lakes are dominated by young carbon. *Nature Clim Change* 8, 166−171 (2018). https://doi.org/10.1038/s41558-017-0066-9

**#24NRH**(Line 389-400): Nice summary paragraph. Kelebar et al. (2023) supports similar interpretations and may be used as additional reference.

Kleber, G.E., Hodson, A.J., Magerl, L. et al. Groundwater springs formed during glacial retreat are a large source of methane in the high Arctic. Nat. Geosci. 16, 597–604 (2023). https://doi.org/10.1038/s41561-023-01210-6

No further comments.

**6 Conclusion**

NRH: The authors present novel data and likely is the first to measure stable carbon isotope signatures of atmospheric methane at hotspots in the Mackenzie River Delta (MRD). The results suggest a variety of methane production sources in the MRD, with source isotope signatures ranging from -42 ‰ (biogenic) to -88 ‰ (thermogenic). Of the eight sites investigated, two had a thermogenic origin, four were biogenic, and two were possibly a result of oxidation of mixed biogenic/thermogenic sources. This is supported by other recent studies.

These results also suggest methane migration from below the thin permafrost at most sites, including from the Taglu gas field, over an area of approximately 20 km north to south, which suggest complex permafrost distribution (e.g. due to the hydrology of river taliks, pingo-systems, coastal settings, and together a mix of likely through-taliks and permafrost degradation).

The study was able to validate airborne eddy covariance hotspot locations using walking transects to measure atmospheric methane variation. However, authors point out these methods only provide a snapshot of methane sources during site visits, and a comprehensive understanding of annual methane production is yet to be established.

Authors suggest that future research should include year-round flux measurements and stable carbon isotope measurements to fully quantify the annual methane emission from both biogenic and thermogenic sources. Additionally, combining portable methane analysers with flux chamber and isotopic measurements could help to better identify and quantify sources, particularly at sites where both biogenic and thermogenic sources are likely. Geophysical mapping atop these transects would provide a useful coupling to permafrost distribution and potential source attribution between biogenic and thermogenic sources.

The study offers valuable insights into the Kohnert et al. (2017) results, which inspired many further airborne and ground measurements across the arctic (e.g. ABoVE domain). Therefore, these results should be published as presented, with additional comments as recommended, and may further enhance these valuable ground truths for current campaigns conducting similar observations.

I highly recommend adding some of the added reference support material (#1-13) and considering comments #1-23. Particularly important, reference material post-Kohnert et al. results from MRD, e.g., Elder et al. (2020, 2021) and Baskaren et al. (2022). I hope these added suggestions and reference support materials can serve to enhance your significant results.

Nice work.

NRH

8-5-2023

End.

**Assignment:** Accepted subject to minor technical corrections or additional reference material, as suggested.

---

## Author Response (AR2)

Anonymous review # 1 comments and reply.

Reviewer # 1,

Thank you for your time and contribution in reviewing this manuscript. All general and main text comments have been updated in the manuscript text as recommended.

Review of 2022-549: "Characterization of atmospheric methane release in the outer Mackenzie River Delta from biogenic and thermogenic sources." by Wesley et al.

Wesley et al. have studied hotspot of methane from the Mackenzie Delta, measuring stable isotopes from atmospheric methane above know aquatic and terrestrial hotspot of methane. Their $\delta^{13}$C-CH4 signatures indicate that both biogenic and thermogenic sources are found in the delta.

I find that study very interesting as methane hotspots are rarely characterized, especially I appreciate the effort of verifying data obtained by airborne eddy covariance. Although the airborne study has been realized in 2013 and the discrete sampling in 2019. In this study they characterized very few hotspots (8) and it would be worthwhile to continue this study and add more data to be able to derive a regional pattern. I recommend for publication with very minor correction.

**General comments:**

Lake 1 is also known under the name "Swiss Cheese lake", it would be useful for the reader not familiar with this area to mention that in the supplement (for example in table S1).

**Main text:**

L168-170: The sentence starting with "Sampling transect locations..."is repeated L231: There is a "-" missing before "53"

Reviewer Nicolas R. Hasson comments and reply:

Dr. Hasson,

Thank you for your time and thorough review of this manuscript. The authors agree that the incorporation of recent (2021- present) literature will strengthen the support for our findings and have decided to incorporate the majority of your suggestions.

**Review:**

"Characterization of atmospheric methane release in the outer Mackenzie River Delta from biogenic and thermogenic sources" (egosphere-2022-549) Submitted on 26 Jun 2022.

**Reviewer:**

Nicholas R. Hasson (NRH), University of Alaska Fairbanks (no-competing conflicts)

**Assessment:**

The study provides significant insight into the findings of Kohnert et al. (2017), which have spurred a multitude of subsequent airborne and ground measurements across the Arctic, a practice that continues to this day. The results suggest a variety of methane production sources in the MRD, with source isotope signatures ranging from -42 ‰ (biogenic) to -88 ‰ (thermogenic), range values indicates that methane in the MRD is likely being produced by both biogenic and thermogenic sources and suggest some mixing, strongly linked to seasonality. This is perfectly in line with what we know about methane production and the interpretation, which seems scientifically valid, does add important reference results to the limited data from airborne/ground coupled surveys on methane hotspot investigations.

I would like the authors to consider recent work (post-Kohnert, 2017) on characterizing hotspots in MRD has provided significant results (e.g. Elder et al., 2020, 2021; Baskaren et al. 2022). Please consider these more-recent updated references on magnitude and occurrence of methane hotspots in MRD, and how this enhances, supports, or disagrees with your findings. Additionally, consider thermogenic and biogenic isotopic signatures from recent similar work (Kleber et al., 2023; Sullivan et al., 2021; Elder et al., 2018), which may support or not the interpretation concluded here. It may be important to briefly mention how so.

However, these results significantly add to import investigation on the discussion on source attribution (e.g. biogenic vs. thermogenic) sources of methane hotspots, particularly important for airborne validation and coupling of ground truth observation, yet data has remained limited. Therefore, this warrant publications with minor technical adjustments, such as the inclusion of additional supporting reference material that supports or challenges these results and/or provides valuable recent observations (post-Kohnert, 2017) that alludes to the behavior of hotspots in MRD. The comments should be viewed as recommended suggestions for these reference materials, and the authors have the discretion to merely incorporate the reference material without my proposed text modifications. However, I have provided some examples of how the text could be altered to either support or challenge interpretations, which may further enrich these crucial ground truths for ongoing campaigns carrying out similar observations.

In summary, these findings are important and significant results for future endeavors focusing on airborne/ground verification of methane hotspots in the Arctic.

**Review Summary:**

The authors present novel data and likely is the first to measure stable carbon isotope signatures of atmospheric methane at hotspots in the Mackenzie River Delta (MRD). The results suggest a variety of methane production sources in the MRD, with source isotope signatures ranging from - 42 ‰ (biogenic) to -88 ‰ (thermogenic). Of the eight sites investigated, two had a thermogenic origin, four were biogenic, and two were possibly a result of oxidation of mixed biogenic/thermogenic sources. This is supported by other recent studies.

These results also suggest methane migration from below the thin permafrost at most sites, including from the Taglu gas field, over an area of approximately 20 km north to south, which suggest complex permafrost distribution (e.g. due to the hydrology of river taliks, pingo- systems, coastal settings, and together a mix of likely through-taliks and permafrost degradation). The study was able to validate airborne eddy covariance hotspot locations using walking transects to measure atmospheric methane variation. However, authors point out these methods only provide a snapshot of methane sources during site visits, and a comprehensive understanding of annual methane production is yet to be established.

Authors suggest that future research should include year-round flux measurements and stable carbon isotope measurements to fully quantify the annual methane emission from both biogenic and thermogenic sources. Additionally, combining portable methane analyzers with flux chamber and isotopic measurements could help to better identify and quantify sources, particularly at sites where both biogenic and thermogenic sources are likely. Geophysical mapping atop these transects would provide a useful coupling to permafrost distribution and potential source attribution between biogenic and thermogenic sources.

I highly recommend adding some of the added reference support material (**#1-13**) and considering comments #1-23. Particularly important, **reference material post-Kohnert et al. results from MRD, e.g., Elder et al. (2020, 2021) and Baskaren et al. (2022).** I would like the **authors to consider (Kleber et al., 2023; Sullivan et al., 2021; Elder et al., 2018**), which may serve to challenge or support these results, and it may be important briefly mentioning how so.

I hope these added suggestions and reference materials can serve to enhance your otherwise significant results. Nice work.

**Review by section:**

**1 Introduction -** Please consider comments/suggestions **#1-11**

**#1**NRH: please consider additional reference supports (A,B), which updates the magnitudes of methane hotspots (including from MRD) and the spatial/temporal distribution, which have been found to follow a power law series as a function of distance to stand water; arctic hotspot methane law (e.g. <40 m from wetland boundary).

**#2**NRH: Furthermore, extreme hotspots have been shown to range from **48 mg to 1008 mg CH4 m$^{-2}$ hr$^{-1}$**, including from MRD region. However, I mentioned from a recent meta analysis on terrestrial sources of methane (e.g. >60 N)(Kuhn et al, 2021) shows the roughly average yearly CH4 emission from

terrestrial permafrost areas to be 2.22 mg m$^{-2}$ hr$^{-1}$. Therefore, your reported {4-5 mg m$^{-2}$ h$^{-1}$ values} **are considered high or roughly double the average hourly mean.**

**#3NRH: I would label extreme hotspots,** including from MRD region, to include the sources found more recently (post-Kohnert study) by Elder et al. (2020,2021) and Baskaren et al. (2022); the work here highlights the importance of your work and significant of the Kohnert et al. follow up (e.g. isotopic fingerprinting associated with hotspots), but currently, lacks these updates references. I've organized potential text below:

A. Additional references for spatial and topographic methane hotspots in MRD

**#4NRH:** I highly recommend adding recent reference support regarding methane hotspot detection in Mackenzie Delta Region (MRD). I've added a potential way to include citation in the text (line 91-95)

Suggested text inclusion
"In the MRD, arctic CH4 hotspots have been found to exhibit a power law relationship with the distance to the nearest standing water (Elder et al., 2020). The geomorphic factors controlling CH4 hotspots in the MRD reveal a spatial decay in the correlation between distance to water and land cover or vegetation type (Baskaran et al., 2022).

Suggested supporting reference #1:
Elder, C. D., Thompson, D. R., Thorpe, A. K., Hanke, P., Walter Anthony, K. M., & Miller, C. E. (2020). Airborne mapping reveals emergent power law of Arctic methane emissions. Geophysical Research Letters, 47, e2019GL085707. https://doi.org/10.1029/2019GL085707

Suggested supporting reference #2:
Baskaran, Latha & Elder, Clayton & Bloom, A. & Ma, Shuang & Thompson, David & Miller, Charles. (2022). Geomorphological patterns of remotely sensed methane hot spots in the Mackenzie Delta, Canada. Environmental Research Letters. 17. https://dx.doi.org/10.1088/1748-9326/ac41fb

Response to Comment 1- 4

The following text has been added to line 45:

"In the MRD, Arctic CH$_4$ the frequency of CH$_4$ hotspots decreases exponentially as distance to standing water increases (Elder et al., 2020; Baskaran et al., 2022)."

The following text has been added at line 78:

"Elder et al. (2021) observed diffusive flux averaging 48.75 mg CH$_4$ m$^{-2}$ hr$^{-1}$ and peaking at 1,008 mg CH$_4$ m$^{-2}$ hr$^{-1}$ directly over thawed permafrost on the edge of a thermokarst lake in interior Alaska."

The following Text has been added to the paragraph from lines 94:

"According to a recent meta-analysis, the cut off value of 5 mg CH4 m$^{-2}$ hr$^{-1}$ used by (Kohnert et al., 2017) is approximately double the mean flux rate for Arctic and boreal regions (Kuhn et al., 2021)".

B. Additional references for methane hotspot magnitudes

**#5**NRH:**:** Please add updated reference to the context of "extreme methane hotspots" equating to {4-5 mg m$^{-2}$ h$^{-1}$}. For example, Elder et al. (2021) reported "Ground-based chamber measurements confirmed average daily CH4 fluxes of 1,170 mg m$^{-2}$ d$^{-1}$, with extreme daily maxima up to 24,200 mg CH4 m$^{-2}$ d$^{-1}$". Converting to hourly, this equates to **48 mg to 1008 mg m$^{-2}$ hr$^{-1}$**. **Which are considered extreme.**

Suggested supporting reference #3:
Elder, C. D., Thompson, D. R., Thorpe, A. K., Chandanpurkar, H. A., Hanke, P. J., Hasson, N., et al.(2021). Characterizing methane emission hotspots from thawing permafrost. Global Biogeochemical Cycles, 35, e2020GB006922. https://doi.org/10.1029/2020GB006922

**#6**NRH: line 94-96 from your text, I recommend adding more recent hotspot values, e.g.,...

... "hotspots identified {were high, when considering the} maximum published value of around 4 mg CH4 m$^{-2}$ hr$^{-1}$ detected for biogenic fluxes north of 61°N (Friborg et al., 2000; Sturtevant et al., 2012; Sachs et al., 2008).

**#7**NRH: <additionally> ...{However, recent hotspot observations from >60°N show more extreme hotspots ranging from **48 mg to 1008 mg CH4 m$^{-2}$ hr$^{-1}$** , including from the MRD (Elder et al., 2020; 2021; Baskaran et al. 2022)}...<additionally>...{and recent meta analysis shows the **yearly daily average of terrestrial hotspots to range 5.04 to 29.3 mg CH4 m$^{-2}$ h$^{-1}$**, when considering other emission inventories >4 mg **CH4 m$^{-2}$ h$^{-1}$**. }. Futhermore, from a more holistic reference gathering, I made a rough calculation of the terrestrial database of methane emissions (Kuhn et al., 2021), including 105 previous methane flux investigations, I calculated the average CH4 emission to be 53.3 mg m$^{-2}$ d$^{-1}$ (range: 705 to -9.8; mg m$^{-2}$ d$^{-1}$ ;N=545). When converting to hourly, the average yearly CH4 emission from terrestrial permafrost areas showed 2.22 mg m$^{-2}$ hr$^{-1}$.

**#8**NRH: Therefore, your reported {4-5 mg m$^{-2}$ h$^{-1}$ values} **are considered high or roughly double the average hourly mean.** However, under the context of eddy covariance open-air mixing values, which are very high in this context (e.g. versus chamber-derived emissions).

**#9**NRH: I would mentioned this. >Values can be higher, with recently reported the yearly daily average of 705 to 121 mg m$^{-2}$ d$^{-1}$ or ranging 5.04 to 29.3 mg m$^{-2}$ h$^{-1}$. Together with the >60°N extreme hotspots ranging from **48 mg to 1008 mg CH4 m$^{-2}$ hr$^{-1}$** , including from the MRD (Elder et al., 2020; 2021; Baskaran et al. 2022)}

Suggested supporting reference #4:

Elder, C. D., Thompson, D. R., Thorpe, A. K., Chandanpurkar, H. A., Hanke, P. J., Hasson, N., et al. (2021). Characterizing methane emission hotspots from thawing permafrost. *Global Biogeochemical Cycles*, 35, e2020GB006922. https://doi.org/10.1029/2020GB006922

Suggested supporting reference #5:

McKenzie Kuhn, Ruth Varner, David Bastviken, Patrick Crill, Sally MacIntyre, et al. 2021. BAWLD-CH4: Methane Fluxes from Boreal and Arctic Ecosystems. Arctic Data Center doi:10.18739/A2DN3ZX1R.

**#10**NRH**:** Note, the isotopic composition of CH4 hotspots characterized in Elder et al. (2021), was later investigated and showed biogenic origin: "**CH4 hotspot revealed a $^{14}$C age of 35,360 YBP ($\delta^{13}$C -73.8 ‰)**(Hasson et al. 2022)." This supports your biogenic source attributions of hotspots detected by airborne observations, although from Alaska.

Response to Comment 5 -10

These are very interesting results by Hasson et al (2022) and Elder et al (2021) that do indeed support the results in this study, they should be mentioned here and further included in the discussion.

The following has been added to line 95:

"More recent work has shown that exceptionally high flux rates averaging 48.75 mg $CH_4$ m$^{-2}$ hr$^{-1}$ can be attributed to biogenic production, with a stable carbon isotope signature of -73.8 ‰ (Elder et al., 2021; Hasson, 2022).

Suggested supporting reference#6

Hasson, N., et al. (2022). Methane emissions show exponential inverse relationship with electrical resistivity from discontinuous permafrost wetlands in Alaska. AGU Fall Meeting Abstracts. AAGUFM.B15E..06H https://ui.adsabs.harvard.edu/abs/2022AGUFM.B15E..06H

**#11**NRH**:** Note, it may also be useful to discuss the Elder et al. (2021) results, which goes on to upscale these observations from arctic and pan-arctic hotspot detection, including MRD:

e.g., "Emissions from the hotspot accounted for ~40% of total diffusive CH4 emissions from the entire study area. Combining these results with hotspot statistics from our 70,000 km$^{-2}$ airborne survey across Alaska and northwestern Canada (e.g. MRD), we estimate that terrestrial thermokarst hotspots currently emit 1.1 (0.1 – 5.2) Tg CH4 yr-1, or roughly 4% of the annual pan- Arctic wetland budget from just 0.01% of the northern permafrost land area." This support MRD hotspot signficance.

Response to comment #11

The following has been added to line 80:

"Terrestrial thermokarst hotspots are estimated to account for roughly 4% of the pan-Arctic $CH_4$ budget but make up only 0.01% of the northern permafrost land area (Elder et al., 2021)"

No further comments.

**Section 2:** Setting - Please consider adding reference material for context/support **#12-13**

**#12**NRH: I recommend the additional references to support your setting, e.g., at line #116-118:

< "Permafrost is generally seen as being continuous under land areas in MRD, but it is mostly missing under lakes that don't freeze all the way to the bottom during winter (Nguyen et al., 2009)." <adding further support context> "Additionally, it has been demonstrated in permafrost regions >60 N, **river taliks extend beyond the river plane by connecting through-taliks with nearby wetlands** (Minsley et al., 2012; see Figs 4,5), showing how "discontinuous" permafrost conditions could be present adjacent to MRD tributaries">

**#13**NRH: **why not include** something like <"The airborne geophysical data detailed in Minsley et al. (2012), illustrates the complexity of river through-taliks and hydrology (e.g. Yukon river) which may mimic areas like the Mackenzie River Delta (MDR). This complexity leads to the presence of discontinuous permafrost near rivers, indicating that the MDR might be fragmented near river tributaries, allowing gas-conduits to form along hydrological through-taliks. These intricate river taliks have the potential to connect with lakes, as observed in the Yukon and Noatak rivers in Alaska, and have been identified in close proximity to some of the largest geological seep sources (Sullivan, 2021). Recent studies have further uncovered that methane-rich groundwater springs are transporting deep thermogenic and biogenic methane gas, such as in Svalbard, north of 79°N latitude. This gas reaches the surface and is found to be supersaturated with methane at levels up to 600,000 times greater than what would be expected for equilibrium with the atmosphere (Klebar et al., 2023).">...

- For example, river talik networks can fragment "continuous permafrost" into "discontinuous permafrost" by through-taliks near large watersheds (e.g. MDR), which may help support why hotspots of deeper thermogenic emissions would occur here in "continuous zone permafrost".

- For example, Sullivan et al. shows the river talik is at work here. and is along the Noatak river, suggesting "microbially produced fossil CH4 is being vented though a narrow thaw conduit below Esieh Lake through pockmarks on the lake bottom. This is one of the highest flux geologic CH4 seep fields known in the terrestrial environment and potentially the highest flux single methane seep",

- For example: recent work (2023) highlights the context of your investigation on thermogenic sources in settings section, e.g., : "

Response to comments 12 and 13

The following has been added to line 114:

"This landscape is a prime location for the formation of Arctic river taliks (Ensom et al., 2012) which can be sources of high-rate geologic CH$_4$ seeps (Sullivan et al., 2021). These river taliks can also form connecting through taliks with nearby wetlands which can create a network of discontinuous permafrost (Minsley et al., 2012)."

Suggested supporting reference #7-9

Minsley, B. J., et al. (2012), Airborne electromagnetic imaging of discontinuous permafrost, Geophys. Res. Lett., 39, L02503, doi:10.1029/2011GL050079.

Sullivan, TD, Parsekian, AD, Sharp, J, et al. Influence of permafrost thaw on an extreme geologic methane seep. Permafrost and Periglac Process. 2021; 32: 484–502. https://doi.org/10.1002/ppp.2114

Kleber, G.E., Hodson, A.J., Magerl, L. et al. Groundwater springs formed during glacial retreat are a large source of methane in the high Arctic. Nat. Geosci. 16, 597–604 (2023). https://doi.org/10.1038/s41561-023-01210-6

No further comments.

**Section 2:** Study Location - Please consider reference support **#14**

**#14**NRH :Line 175 reference suggestion (context)...e.g., **context for the geophysical extent of through-taliks near sites (1) channel seep, Pingo 2, Pingo 1, and Site 9...**

>"However, nearby reports along the arctic coastal shelf have indicated the lack of ice-bonded permafrost beneath the coastline, implying the existence of extensive through-taliks that intrude sea-to-land. These through-taliks are potentially connecting subpermafrost aquifers on land, alluding to the poorly understood and highlight complex dynamics of coastal subsurface hydrology."> impacting potentially (1) channel seep, Pingo 2, Pingo 1, and Site 9...

Suggested supporting reference #10
Micaela N. Pedrazas et al., Absence of ice-bonded permafrost beneath an Arctic lagoon revealed by electrical geophysics.Sci. Adv.6,eabb5083(2020).DOI:10.1126/sciadv.abb5083

Response to comment 13:

The authors do not believe that this reference is good support for through taliks at the near shore sites.

No further comments.

**Section 3:** Methods, Sample collection and analysis - comments/suggestions **#14-15**

Please consider added reference to warrant justification of sampling protocols on hotspots, which support authors approach, which does support initial hypothesis (e.g. distance from standing water=larger source attribution target).

NRH: The authors used similar method is based on protocols by Andersen et al., 2018. Authors point out that this method are only able to recover a small fraction of the hotspot (e.g. 5-10% by area). Note, as previous aforementioned studies allude to (e.g. Elder et al., 2021, Hasson et al., 2022), even though hotspots can dominate the local diffusive CH4 budget (e.g. 90% total), these arise from only a tiny fraction of the area (e.g. 15% of observations), suggesting that although Kohnert et al. observations show large coverage (e.g. 1 km2), these hotspots may come from discrete areas (e.g. 150-200 m transects), suggesting this study may have captured these disproportionally large sources from a disproportionally small area, similar to Elder et al. 2021

**#14**NRH: The importance of these sampling protocols rest on the hypothesis that the sample locations chosen (e.g. near wetlands and pingos, etc.) are the source of these hotspots. **This is not necessarily unwarranted bias,** since prior work has shown that hotspots follow a power law series from distance to standing water (e.g. 0-100 m)(e.g. Elder et al. 2020, 2021; Baskaran et al. (2022)). Its interesting that the hotspots from pingos are near the lowlands (e.g. supporting Baskaran et al., analysis of topographical controls). Given the considerable expensive of field work by helicopter and challenging environment,

which limits sampling observational time, future work may also consider transects <100 m from wetlands or areas with complex through-taliks and complex permafrost hydrology (e.g. Pingos, tributaries, coastline).

The authors may wish to highlight this fact for MRD hotspots to minimize effects of bias or rational for the interpretation of data.

Therefore, if sampling protocols reference the power law findings by Elder et al., 2020,Baskaran et al. 2022, the bias near wetlands has a justified reason. Furthermore, discrete changes from upland to lowlands (escarpment bluffs, river terraces, lake terraces, etc.) have been shown to be statistically related to high occurrence of MRD hotspots (Baskaran et al. (2022)). **The authors sampling protocol near distance to standing water or topographic changes near targets is a good approach.**

Response to Comment 14:

The authors agree that this is not an unwarranted bias but would like to be clear to the reader that they had to make a choice when sampling that could have biased the results to sources from wetlands and pingos.

**#15**NRH: Suggestion, it might be advantages to show the sampling transects as a function of distance to standing water (e.g. Channel seep was ~50 m from standing water, etc.). Does this spatial relationship tell us anything more about the data (e.g. isotopic signature, concentrations?). Perhaps the authors can mention any relationship that is found, e.g. similar to the pingo lowland having higher concentration than pingo upland, so future studies can plan sampling around these known spatial relationships between geomorphology and water table position.

Response to Comment 15

There was a relationship as the 2 definitive thermogenic sites are both aquatic seeps, increasing the possibility of through taliks. The authors believe this is adequately addressed in the discussion.

No additional comments on methods here.

**Section 3:** Methods, Determination of CH4 source stable isotope value - **#16-17**

NRH: Authors justify mass balance approach, considering limitations and assumptions. In this context, sampling minimized bias by sampling upwind and downwind of the estimated locations, along 600-800 m transects, which provided a large spatial and temporal range estimate of source attributions. Wind regime was observed using handled anemometer with accuracy of +/- 0.1 m/s. From the paragraph (L239-256), the authors acknowledge the limitations of the Keeling plot analysis under certain field conditions. They accept that the method's assumption of only two components being measured (the source released at the surface/atmosphere interface and the background regional atmospheric signature) can be challenging in broad areas or windy conditions that may cause mixing.

Given these constraints, its acceptable to assume these limitations when applying the Keeling plot analysis, which depend on the specific context of the hypothesize: the authors clarify earlier that hotspots may form from discrete areas near wetlands and pingos. It seems than the significance of the data is determined by the wind direction from source target (wind direction from or away source target). The authors have acknowledged the limitations and adapted their methodology, accordingly, attempting to collect samples as close to the known point source of emissions as possible when observable ebullition was present.

Their acceptance of the limitation when appraising large hotspots, based on their assumption that the atmospheric point samples taken within these hotspot source regions could represent a mixed δ13C-CH4 signature, seems a reasonable approach given the constraints of field conditions and the necessity to identify broad trends.

**#16**NRH: I recommend complementary rose wind plots that show the wind direction versus source target over time, and how that may contribute to the interpretations of results. Showing wind rose plots (over time, e.g. 30 minutes) shows how the upwind effects total wind mixing effects on dilute

concentrations or may allude to the source (e.g. increased as wind shifted NE, versus SW). Although, currently, the authors do show in Figure 3 the wind direction (red arrow vector), **but perhaps a wind rose representing the time domain of sampling could be shown.** Although, not necessary, if this "red arrow" on wind represents a mean over the duration of time sampled or constant direction.

**#17**NRH: **Does the red vector arrow show the mean or constant direction of wind over the sampling period?** If so, mentioned this in Figure 3 caption (e.g. wind direction represents average during sampling time).

Response to Comment 16 and 17

The following has been added to Figure 3 Caption: "The red vector arrow indicates the constant wind direction during the sampling period."

No further comments.

**Section 4**: Results

NRH: In summary, the results show trends of elevated methane and carbon dioxide concentrations in specific sites and a variability in stable carbon isotope signatures, indicating possible differences in the sources of these greenhouse gasses at each site. The results highlight the observation of elevated atmospheric CH4 concentrations at four of the five (Pingo 1, Pingo 2, Wetland 2, Wetland 3) where walking transects were performed.

The results (as expected, from hypothesis) show that proximity to source of seep, e.g., ebullition (closest to standing water) which had substantially elevation methane concentrations. From this, I am curious of the relationship between distance to standing water and/or topographic upland versus lowland relationship to both concetrations and isotopic signatures elsewhere. Also noteworthy, Keeling plot values show -88.4‰ for wetlands in the fall, during maximum thaw, or when thermal lag from summer months penetrates deeper in talik sources, given the latency heat of water. Whereas Keeling plot values show -56.7‰ during summer months, when the cold season thermal lag can be substantial in the subsurface.

**#18**NRH: Perhaps this suggest age dependency on talik versus time of thermal lag in subsurface and seasonal thermal lags associated with age. Alternatively, hydrology or seasonality of groundwater surge (higher in summer, lower in august) may result in more mixing effects. It might be useful to discuss this later.

Line 273-274: > Keeling plot values were -88.4‰ for Wetland 1 when sampled in the fall, but - 56.7 ‰ when sampled in the summer.>

**#19**NRH: Please add month (x), e.g., fall (X), summer (X) in Line 273-274. No further comments.

Response to Comment 18 and 19

"October" and "July" have been added to line 247

**Section 5.1:** Results - Consider comments on reference material #20-21 - Outer delta pingos

NRH: The study discovered high methane concentrations in airborne hotspots in the northwestern outer MRD. The carbon isotope signatures from these sites suggest a likely mixture of thermogenic and biogenic methane sources. Despite the proximity of these high readings to pingo features, notably, **the highest values were detected in the surrounding low-lying shrub tundra, not the features themselves.**

**#20**NRH: This suggest topographical relationships to hotspots or water table position by Elder et al., Baskaren et al., etc.

NRH (Line 315): indeed, I agree, this is exciting result and assumptions about the permafrost through-taliks, pingo hydrology, and complex permafrost and absence of permafrost does warrant geophysical transects. It seems understanding the source attribution is a function of the assumptions about permafrost or absence of permafrost, which can be demarcated by low- frequency EM geophysical transects.

Response to Comment 20:

The following text has been added to line 282:

"This interpretation would be consistent with recent findings in the region that water table depth, hydrology and topography are critical factors driving emissions at hotspots in the MRD (Elder et al., 2020; Baskaran et al., 2022; Hodson et al., 2020a, b)."

- Wetland sites

<sampling at Wetland 1 revealed a seasonal variability in the carbon isotope signature, with a value of -88.3‰ in October (suggesting a biogenic source) and -53.4‰ during the summer (indicating a mixed source). This could be due to methane oxidation or a blend of biogenic and thermogenic sources.> >The lack of observed methane ebullition during summer suggests a potential seasonal variation in methane flux in these wetland settings.>Therefore, contributions to the atmospheric methane at Wetland 1 during summer could be from both biogenic and thermogenic sources, with oxidation and varying production pathways also being plausible.>

>Similar observations for seasonal variability in terrestrial sources are not well documented in the literature, although transport of CH4 from anaerobic soils with sedge vegetation has been observed to bypass the aerobic zone, limiting oxidation during the growing season (Olefeldt et al., 2013; King et al., 1998). Therefore, it is possible that there were contributions to the atmosphere from biogenic and thermogenic sources at Wetland 1, but oxidation and varying production pathways cannot be ruled out as the reason for the signature derived during the summer sampling>

**#21**NRH:Interesting result. This may also allude to thermal lag times moving through different depths seasonally or perhaps more important here, the mixing blend of biogenic and thermogenic sources are

enhanced by summer peak water levels, which can transport various methane sources significantly in the watershed. **Groundwater discharge as a driver of methane emissions and mixing from Arctic lakes and transporting methane from upland active layer thaw to lowland area has been shown to be significant,** supporting oxidation and blending of biogenic and thermogenic source.

... **Ground water discharge is known to follow as seasonal trend, which perhaps alludes to the shifts from summer to fall data, with greater mixing of biogenic and thermogenic sources in summer, and more stable (deeper) biogenic sources emanating directly from the talik, without ground water mixing.** For example, Please see Olid et al( 2022) and Paytan et al. (2015).

Suggested supporting reference #11

Olid, C., Rodellas, V., Rocher-Ros, G. et al. Groundwater discharge as a driver of methane emissions from Arctic lakes. Nat Commun 13, 3667 (2022). https://doi.org/10.1038/s41467-022-31219-1

Suggested supporting reference #12
Lecher, Alanna & Dimova, Natasha & Sparrow, Katy & Garcia-Tigreros, Fenix & Murray, Joseph & Tulaczyk, Slawek & Kessler, John. (2015). Methane transport from the active layer to lakes in the Arctic using Toolik Lake, Alaska, as a case study. Proceedings of the National Academy of Sciences of the United States of America. 112. 10.1073/pnas.1417392112.

Response to Comment 21:

The following has been added to line 315:

"Groundwater inputs to lakes in permafrost areas are higher during the summer months (Olid et al., 2022), which would increase the possibility of thermogenic inputs during the summer."

No further comments.

**Section 5.2:** Results - Consider more recent reference support #21 and counter evidence #22,23 (if applicable), and further recent support for thermogenic versus biogenic signatures in arctic from #24

Line 368: >Some of the largest occurrences of atmospheric release of CH4 in Arctic environments have been reported in association with large gas seeps of thermogenic CH4, causing high-rate ebullition (Walter Anthony et al., 2012)>

**#21**NRH: consider also adding the more recent literature for support for both biogenic sourced geological gas and thermogenic >

Kleber, G.E., Hodson, A.J., Magerl, L. et al. Groundwater springs formed during glacial retreat are a large source of methane in the high Arctic. Nat. Geosci. 16, 597–604 (2023). https://doi.org/10.1038/s41561-023-01210-6

Sullivan, TD, Parsekian, AD, Sharp, J, et al. Influence of permafrost thaw on an extreme geologic methane seep. Permafrost and Periglac Process. 2021; 32: 484–502. https://doi.org/10.1002/ppp.2114

**#22**NRH: I am curious how these results of δ13C-CH4 source signatures (e.g. -42.0 to 44.7 ‰) challange large gas seeps of thermogenic CH4, when Elder et al. (2018) showed these signatures associated with alaskan lakes dominated by young carbon, which may challenge this interpretation.

For example, you may wish to add this edits:

<Some of the largest occurrences of atmospheric release of CH4 in Arctic environments have been reported in association with large gas seeps of thermogenic CH4, causing high-rate ebullition (Walter Anthony et al., 2012). **However, arctic coastal plain CH4 emissions have been associated with younger carbon sources (Elder et al., 2018)**">

Response to Comment 21 and 22:

The following has been added at line 327:

"but biogenic contributions from microbially produced fossil $CH_4$ (Sullivan et al., 2021) and young carbon can also occur at seeps with high flux rates (Elder et al., 2018)"

**#23**NRH: you may want to download Elder et al. supp section and take a look at those corresponding δ13C-CH4 signatures and similarities or dissimilarities to your data.

Suggested supporting reference #13

Elder, C.D., Xu, X., Walker, J. et al. Greenhouse gas emissions from diverse Arctic Alaskan lakes are dominated by young carbon. Nature Clim Change 8, 166–171 (2018). https://doi.org/10.1038/s41558-017-0066-9

**#24NRH** (Line 389-400): Nice summary paragraph. Kelebar et al. (2023) supports similar interpretations and may be used as additional reference.

Kleber, G.E., Hodson, A.J., Magerl, L. et al. Groundwater springs formed during glacial retreat are a large source of methane in the high Arctic. Nat. Geosci. 16, 597–604 (2023). https://doi.org/10.1038/s41561-023-01210-6

Response to Comment 23 and 24:

The following has been added at line 353:

"Stable carbon isotope values even higher (more enriched in $^{13}C$ ) than those reported in this study (-38.8 ‰) were observed below lake ice on the north slope of Alaska (Elder et al., 2018). These higher values were attributed to oxidation of biogenic $CH_4$ but, were measured in an environment where high rates of oxidation are likely."

No further comments.

**6 Conclusion**

NRH: The authors present novel data and likely is the first to measure stable carbon isotope signatures of atmospheric methane at hotspots in the Mackenzie River Delta (MRD). The results suggest a variety of

methane production sources in the MRD, with source isotope signatures ranging from -42 ‰ (biogenic) to -88 ‰ (thermogenic). Of the eight sites investigated, two had a thermogenic origin, four were biogenic, and two were possibly a result of oxidation of mixed biogenic/thermogenic sources. This is supported by other recent studies.

These results also suggest methane migration from below the thin permafrost at most sites, including from the Taglu gas field, over an area of approximately 20 km north to south, which suggest complex permafrost distribution (e.g. due to the hydrology of river taliks, pingo- systems, coastal settings, and together a mix of likely through-taliks and permafrost degradation).

The study was able to validate airborne eddy covariance hotspot locations using walking transects to measure atmospheric methane variation. However, authors point out these methods only provide a snapshot of methane sources during site visits, and a comprehensive understanding of annual methane production is yet to be established.

Authors suggest that future research should include year-round flux measurements and stable carbon isotope measurements to fully quantify the annual methane emission from both biogenic and thermogenic sources. Additionally, combining portable methane analysers with flux chamber and isotopic measurements could help to better identify and quantify sources, particularly at sites where both biogenic and thermogenic sources are likely. Geophysical mapping atop these transects would provide a useful coupling to permafrost distribution and potential source attribution between biogenic and thermogenic sources.

The study offers valuable insights into the Kohnert et al. (2017) results, which inspired many further airborne and ground measurements across the arctic (e.g. ABoVE domain). Therefore, these results should be published as presented, with additional comments as recommended, and may further enhance these valuable ground truths for current campaigns conducting similar observations.

I highly recommend adding some of the added reference support material (#1-13) and considering comments #1-23. Particularly important, reference material post-Kohnert et al. results from MRD, e.g., Elder et al. (2020, 2021) and Baskaren et al. (2022). I hope these added suggestions and reference support materials can serve to enhance your significant results.

Nice work. NRH 8-5-2023 End.

**Assignment:** Accepted subject to minor technical corrections or additional reference material, as suggested.

**References:**

Baskaran, L., Elder, C., Bloom, A. A., Ma, S., Thompson, D., & Miller, C. E. (2022). Geomorphological patterns of remotely sensed methane hot spots in the Mackenzie Delta, Canada. *Environmental Research Letters*, *17*(1), 015009. https://doi.org/10.1088/1748-9326/ac41fb

Elder, C. D., Thompson, D. R., Thorpe, A. K., Chandanpurkar, H. A., Hanke, P. J., Hasson, N., James, S. R., Minsley, B. J., Pastick, N. J., Olefeldt, D., Walter Anthony, K. M., & Miller, C. E. (2021). Characterizing Methane Emission Hotspots From Thawing Permafrost. *Global Biogeochemical Cycles*, *35*(12). https://doi.org/10.1029/2020GB006922

Elder, C. D., Thompson, D. R., Thorpe, A. K., Hanke, P., Anthony, K. M. W., & Miller, C. E. (2020). Airborne Mapping Reveals Emergent Power Law of Arctic Methane Emissions. *Geophysical Research Letters*, 10.

Elder, C. D., Xu, X., Walker, J., Schnell, J. L., Hinkel, K. M., Townsend-Small, A., Arp, C. D., Pohlman, J. W., Gaglioti, B. V., & Czimczik, C. I. (2018). Greenhouse gas emissions from diverse Arctic Alaskan lakes are dominated by young carbon. *Nature Climate Change*, *8*(2), 166–171. https://doi.org/10.1038/s41558-017-0066-9

Hasson, N. (2022). *Methane emissions show exponential inverse relationship with electrical resistivity from discontinuous permafrost wetlands in Alaska*. AGU Fall Meeting. https://ui.adsabs.harvard.edu/abs/2022AGUFM.B15E..06H

Kohnert, K., Serafimovich, A., Metzger, S., Hartmann, J., & Sachs, T. (2017). Strong geologic methane emissions from discontinuous terrestrial permafrost in the Mackenzie Delta, Canada. *Scientific Reports*, *7*(1), 5828. https://doi.org/10.1038/s41598-017-05783-2

Kuhn, M. A., Varner, R. K., Bastviken, D., Crill, P., MacIntyre, S., Turetsky, M., Walter Anthony, K., McGuire, A. D., & Olefeldt, D. (2021). BAWLD-CH$_4$: A comprehensive dataset of methane fluxes from boreal and arctic ecosystems. *Earth System Science Data*, *13*(11), 5151–5189. https://doi.org/10.5194/essd-13-5151-2021

Minsley, B. J., Abraham, J. D., Smith, B. D., Cannia, J. C., Voss, C. I., Jorgenson, M. T., Walvoord, M. A., Wylie, B. K., Anderson, L., Ball, L. B., Deszcz-Pan, M., Wellman, T. P., & Ager, T. A. (2012). Airborne electromagnetic imaging of discontinuous permafrost: AEM IMAGING OF PERMAFROST. *Geophysical Research Letters*, *39*(2), n/a-n/a. https://doi.org/10.1029/2011GL050079

Sullivan, T. D., Parsekian, A. D., Sharp, J., Hanke, P. J., Thalasso, F., Shapley, M., Engram, M., & Walter Anthony, K. (2021). Influence of permafrost thaw on an extreme geologic methane seep. *Permafrost and Periglacial Processes*, *32*(3), 484–502. https://doi.org/10.1002/ppp.2114